



# An assessment of the variability in temperature and salinity of the Baltic Sea from a simulation with data assimilation for the period 1990 to 2020

Ye Liu[1], Lars Axell[1], and Jun She[2]

[1]Swedish Meteorological and Hydrological Institute, Norrköping 60176, Sweden.
[2]Danish Meteorological Institute, Copenhagen 2100, Denmark.

*Correspondence to*: Ye Liu (ye.liu@smhi.se)

**Abstract.** A Baltic dataset covering 1990–2020 is reconstructed using a circulation model and data assimilation. Satellite observations of sea surface temperature and temperature and salinity (T/S) profiles are used to reduce model biases by a

local Singular Evolutive Interpolated Kalman filter. The dataset is evaluated with assimilated T/S profiles and reprocessed grid observations, and the results demonstrate that the sea surface temperature, sea surface height, mixed layer depth, and vertical distribution of T/S are all reasonably reproduced. T/S trends at various depths in the Baltic sub-basins are analyzed from a reanalysis perspective, revealing a clear warming trend in recent decades, with a slight desalination trend in the northern Baltic Sea and a salination trend in the southern Baltic Sea. In particular, T/S trends of the Baltic Sea are larger in

the south than in the north. In the Baltic Sea over the past 30 years, the temperature rises at a rate of 0.036 to 0.041 ºC/year, with a larger warming trend below the thermocline than above it, while the salinity increases with a trend of -0.0036 to 0.049 PSU/year. In addition, seasonal variations are evident in the temperature at the surface, 60 m, and bottom, as well as in the surface salinity, whereas no clear seasonal variations are detected in the salinity below the surface and temperature at 100 m.

## 1. Introduction

The Baltic Sea variability has received considerable attention from both the scientific community and the political and economical communities due to its significant impact on the Nordic climate change. In particular, the Baltic Sea warmed faster than any other coastal sea during the period 1982–2006 (Meier et al. 2017; Belkin 2009). Furthermore, the Baltic Sea is projected to have warmer water temperatures and less sea ice cover by the end of this century (Omstedt et al., 2012). The Baltic Sea exchanges with the North Sea through the Danish Strait in the transition zone. The renewal of the deep water

below the permanent halocline largely depends on the brackish inflows from the North Sea. According to the available observation record, small and medium-sized inflows are relatively frequent, while Major Baltic Inflows (MBI) are irregular and typically weather-driven events (Matthäus et al. 2008). However, the upper-water hydrography is remarkably influenced by river discharge (Bergström and Carlsson, 1994) and net precipitation (precipitation minus evaporation).



Observations in recent decades have revealed an accelerating trend in the warming of the Baltic Sea. The warming rate

of the Baltic sea surface temperature (SST) in summer is three times the global rate since 1985 (Mackenzie and Schiedek, 2007). The Baltic Sea exhibits various features over time due to local changes in climate and forcing, such as an SST increase of 0.05 °C/decade during 1856–2005 and 0.4 °C/decade during 1978–2007(Kniebusch et al, 2019). Therefore, accurate estimation of the past changes in the Baltic Sea is crucial, not only for understanding climate dynamics but also for managing the Baltic marine ecosystem (Belkin, 2009; Reusch et al, 2020; Bonsdorff, 2021; Viitasalo et al, 2022). Although

several studies have focused on long-term changes in the Baltic Sea using observations over the last two decades (Fonselius and Valderrama, 2003; Winsor et al., 2001), a comprehensive climate assessment directly from the observations is still lacking due to the spatiotemporal coverage limitations, especially in the deep Baltic Sea. A reliable data reconstruction provides an opportunity to estimate climate variability and associated uncertainties more comprehensively, and this approach has received increasing attention in the Baltic Sea (Madsen et al., 2015; Høyer and Karagali, 2016; Madsen et al., 2019).

However, a reliable data reconstruction typically requires a certain number of observations, and in the Baltic Sea, only surface data such as SST and sea level are typically reconstructed directly from observations due to improved observation coverage by satellite networks. To overcome limitations in observation coverage, hydrographic climatology is used directly as the background for objective analysis, with or without hydrographic observation contribution (Good et al., 2013). However, this approach does not consider local temporal variations in the ocean state, making it unsuitable for analysing

trends. Numerical simulations have also been used to assess long-term Baltic variability (Meier and Kauker, 2003; Liu et al., 2017), but the circulation models used in these simulations often fail to accurately capture the potential circulation responses to temperature/salinity variability due to imperfect initial conditions and biased forcing (Liu et al. 2013, 2014).

To overcome the limitations of observation coverage and numerical models, reanalysis provides an alternative way that can combine the accuracy of observations with the spatiotemporal coverage of dynamic numerical models. Currently,

several ocean reanalyses are currently available for the Baltic Sea, including BSRA-15, a 15-year reanalysis using the successive corrections method for the period 1990–2004 (Axell, 2013); a 20-year (1990-2009) reanalysis based on the 3DVAR and OI method (Fu et al. 2012); a 30-year reanalysis covering 1970–1999 using EnOI (Liu et al. 2013, 2017); a reanalysis from 1989 to 2013 using the 3D Ensemble Variational method (Axell and Liu, 2016); the coupled physical and biogeochemical reanalysis by Raudsepp et al. (2019), and the latest Baltic Sea Physics Reanalysis (2023) covering 1993-

2021. These reanalyses have been successfully used to validate ocean models, provide initial and boundary conditions for regional simulations and climate analysis, and have been shown to be effective in improving the accuracy of these models (Liu et al., 2014, 2017; Meier et al. 2019). The Baltic Sea Marine Monitoring Forecasting Centre has recently updated the circulation model for the Baltic Sea as part of the Copernicus Marine Environment Monitoring Service (CMEMS), and the observation coverage in the Baltic Sea has greatly improved since 1970, along with the development of the advanced Data

Assimilation (DA) systems. This study aims to provide a quantitative assessment of a physical reanalysis of the Baltic Sea for the period 1990–2020, estimating long-term and seasonal variations of temperature and salinity (T/S) in the Baltic sub-



basins. This is the first attempt to estimate the T/S variability at various depths within the Baltic sub-basins from a reanalysis perspective.

In this study, most of the available T/S observations were assimilated into a state-of-the-art circulation model of the
Baltic Sea. The Local Singular Evolutive Interpolated Kalman (LSEIK) filter (Pham, 1998) was utilized to combine the information from the model and several reliable measurement datasets which includes the satellite sea surface temperatures (SSTs) and the T/S profiles in the Baltic Sea.

## 2. Ocean circulation model

The circulation model called NEMO-Nordic is used to simulate the hydrographic dynamics of the Baltic Sea (see
Figure 1, Hordoir et al., 2019). The model in this study uses a version with a horizontal resolution of 2 nautical miles (~ 3.7 km) and 56 unevenly distributed vertical levels. There is a vertical resolution of 3 m from the surface to the typical halocline level at 60 m. The layer thickness increases significantly below 60 m: the layer thickness is 10m at a depth of 100 m and maximum values of 22 m are reached at the bottom of the deepest part of the domain (Norwegian trench). NEMO-Nordic locates its open boundaries in the English Channel and between Scotland and Norway. At the boundaries, the barotropic
mode is defined by the Flather scheme (Flather, 1994), while the flow relaxation scheme (Davies, 1976) is used for the tracers. In addition, a free-slip option is used for lateral boundaries. At the surface, a bulk formulation based on Large and Yeager (2004) is applied for the surface boundary condition. A nonlinear free surface formulation (Adcroft and Campin, 2004) is adopted with a time-splitting approach to compute the barotropic and baroclinic modes. This study uses the EOS-80 equation to define seawater properties. The modeled water temperature and salinity are in the potential temperature (degrees
Celcius) and Practical Salinity Units (PSU). When comparing against observations, the model's temperature is converted to in situ temperature. The LIM3 model (Vancoppenolle et al., 2008) is coupled with NEMO-Nordic to simulate the sea ice. NEMO-Nordic uses a total variation diminishing (TVD) advection scheme with a modified leapfrog approach to ensure a very high degree of tracer conservation (Leclair and Madec, 2009). The κ-ε scheme is used to parameterize the unresolved vertical turbulence (Umlauf and Burchard, 2003). Galperin parameterization is applied for producing a stable Baltic
stratification (Galperin et al., 1988). The model simulated the horizontal diffusion of momentum and tracers by a Laplacian diffusion with constant-in-time diffusivity and viscosity. A good compromise is to set the diffusivity to 10% of the viscosity value, which allows the model to be barotropically and baroclinically stable while ensuring low viscosity and diffusivity. Further, a strong isopycnal diffusion is used close to the Neva river inflow. At the bottom, quadratic friction is applied with a constant bottom roughness of 3 cm, and the drag coefficient is computed for each bottom grid cell.  In addition, the bottom
boundary layer is parameterized to ease the propagation of saltwater inflows between the Danish Straits and the deepest layers of the Baltic Sea (Beckmann and Doscher, 1997).

The spin-up run (1 January 1975 – 31 December 1989) was initialized by model data from Hordoir et al., (2019). The model is forced by atmospheric forcing from the UERRA reanalysis product with an 11 km and 1-hour resolution (Niermann





et al. 2017). The precipitation from UERRA was corrected by a multiplier of 0.8. At the boundaries in the North Sea, the
ECMWF ORAS4/5 reanalysis data (Balmaseda et al. 2013; Zuo et al., 2018) with a northeast Atlantic barotropic surge
model (NOAMOD; She et al., 2007b) is used to provide the wind-driven sea level, depth-averaged velocity, and vertical
profiles of temperature and salinity. Daily river discharge is taken from a dataset by the EHYPE model (Arheimer et al.,
2012). The salinity of the riverine water is set to a constant value of $10^{-3}$ PSU, which is the same value used for the sea ice at
the river mouths. The river temperature is equal to the sea surface temperature at the same horizontal coordinate.

## 3. Methodology

### 3.1 Data assimilation

The LSEIK filter is adopted to correct the NEMO-Nordic simulation by weighting the difference between
observations and the model state prediction (Pham et al., 1998). The weight coefficients are determined by the error
covariances of the model and the observations. When the observation is available, the LSEIK filter merged the information
from model and observations to produce the analysis state with the formula:

$$X^a = X^f + K\left[Y^o - HX^f\right] \tag{1},$$

$$K = P^f H^T \left(H P^f H^T + R\right)^{-1} = LU(HL)^T R^{-1} \tag{2},$$

$$U^{-1} = \rho r T^T T + L^T H^T R^{-1} HL \tag{3},$$

$$L = X^f T \tag{4}.$$

Here, the superscripts "f", "a", "o", and "T" denote the forecast, analysis, observation, and the transpose of matrix,
respectively. The analysis state, denoted by X, consists of temperature, salinity, and horizontal velocity. The observations are
represented by Y, and H is the linearization of the observation operator that maps the model space to the observation space.
R represents the observation error covariance matrix, while r is the minimum number of sample ensemble members for the
error covariance matrix. The inflation factor is denoted by $\rho$. The matrix T has a dimension of r × (r -1) and has zero column
sums. In the LSEIK analysis, the uncertainty of the model state is estimated using the spread of the sample ensemble. To
remove the unrealistic long-range correlations, horizontal localization is performed with a uniform influence radius of 70 km
(Liu and Fu, 2018; Liu et al. 2013). No vertical localization is applied. Within the radius, observations are weighed by a
quasi-Gaussian function based on the distance between the analyzed grid point and the observation location (Liu et al.
2013). It should be note that this localization length value may not be perfect for LSEIK assimilation in the Baltic Sea.
Furthermore, this research implemented a fixed inflation factor (ρ = 0.3) as Liu and Fu (2018), and used a univariate
updating scheme with the DA increment to achieve the final analysis state.

In this study, the LSEIK is implemented offline, meaning that the ensemble samples are not integrated forward to
predict the model errors. The model uncertainty is estimated from an ensemble of model states without DA. First, a sample
pool was first constructed by storing a snapshot every 3 days from a free run covering the period 1990-2016. A bandpass



filter was applied to remove unexpected high and low frequency variability in the model. Then, 20 samples were selected from each year in a consecutive 6-year centered on the analysis time in a 90-day time window, resulting in a total of 120 samples to estimate the model variability. A multivariable empirical orthogonal function (EOF) decomposition was performed for the covariance matrix of this selected sample ensemble. A second-order exact sampling method with those leading EOF modes is used to generate the sample ensemble (Liu and Fu, 2018). This study applied a 72-hour assimilation

window to yield the "new" initial condition. At every assimilation time, all observations from the previous day to the following day were selected. To reduce the computational cost, the sample ensemble was updated every two assimilation cycles.

   Furthermore, although our model has a horizontal resolution of 2 nm, the mask coordination for land and ocean may be inconsistent between the observations and the model. This inconsistency could result in the wet point defined by the

observations being at the land point defined by the model coordinates. Large adjustments by DA corresponding to large discrepancies between the model and the observations could break the dynamic balance of the model, resulting in unstable simulations. To obtain high quality DA results, quality controls were applied before the observations were used in an assimilation. In the control, a T/S observation was excluded if it deviated from the prediction by more than a threshold of 3 ℃ or 3 PSU, respectively. Observations located on the NEMO-Nordic land grid were also eliminated and duplicate

observations were removed. At the model level, the average of multiple observations was used instead of individual observations to represent the general measure for the level.

   The definition of observation error is commonly constructed based on different assumptions (Liu et al., 2009; Xie and Zhu, 2010; Axell and Liu, 2016). In this study, DA assumes uncorrelated observation errors, and the observation error is estimated to be one value as the sum of all observation uncertainties (Liu et al. 2013). Thus, the observation error covariance

is a diagonal matrix where the diagonal elements are the observation error variance. An exact observation error is almost impossible to obtain. Following She et al. (2007a), the standard deviation (SD) of the satellite SST was defined as a constant value of 0.3 ℃ for all positions of the Baltic Sea. The error variances of the profiles for T/S were designed to increase with the observation depth according to the following equations:

$$\begin{cases} \sigma^2 = \varepsilon_{obs}^2 & d \leq 70m \\ \sigma^2 = \varepsilon_{obs}^2 \times exp(-0.005 \times d) & d > 70m \end{cases} \tag{5}.$$

Here $d$ is the observation depth. $\varepsilon_{obs}$ is the predefined observation error, which equals 0.5 ℃ for a temperature profile and 0.12 PSU for a salinity profile.

### 3.2. Observations

  In this study, remote sensing SSTs and T/S profiles were assimilated into NEMO-Nordic. Observations from satellite and in-situ were used to validate the reconstructed dataset.



### 3.2.1 The satellite observations


The first satellite dataset assimilated in this study consists of SST observations obtained from the Ocean and Sea Ice Satellite Application Facility (OSISAF, 2022). This dataset derives from the subskin temperature for the ocean of 50−90oN at a resolution of 5 km (Donlon et al. 2012). To minimize the influence of diurnal warming effects in the Baltic Sea, only night-time measurements were used during the reconstruction process (Karagali et al. 2012; Karagali and Høyer 2014).

Additionally, only high-quality observations that passed the climatology check were included in this study. The second dataset used in this study is the SST measurements obtained from Swedish ice service digitized sea-ice chart data (IceMap, 2022). This dataset has a spatial resolution of 5 km and is manually quality checked. During the ice season, this product is updated almost daily in 1998, but two or three times per week in other years. The reconstructed SST was validated using the CMEMS level-4 SST (CMEMS_L4, Høyer and Karagali, 2016). The CMEMS_L4 SST observations are available daily with

a spatial resolution of 0.03o, which is a gap-free observation record reconstructed by combining the Advanced Very High Resolution Radiometer (AVHRR) Pathfinder dataset and the Along-Track Scanning Radiometer (ATSR) Reprocessing for Climate dataset.

### 3.2.2 In situ data

The T/S profiles assimilated in this study were collected from the Swedish Oceanographic Data Centre (SHARK, 2022)

and the International Council for the Exploration of the Sea (ICES, 2022). These datasets mainly contain low-resolution Conductivity-Temperature-Depth (CTD) sensors and bottle data at predefined standard stations in the Baltic Sea, as well as connecting regions between the North Sea and the Baltic sea (e.g. Kattegat and Skagerrak, see Fig.1). The observations in the SHARK and ICES data center were checked for location and local stability. Sea level observations at the tide gauge stations were also downloaded from Baltic Sea - near real-time (NRT) in situ quality-controlled observations (2022) to

evaluate the reconstructed results. Figure 2 shows the spatio-temporal distribution of the observed T/S profiles used in the reanalysis for the period 1990-2020. The number of profiles used in this study is clearly inhomogeneous in time and space, with the maximum number of observed profiles for T/S being 4747 in 2016. The number of observed profiles increases significantly with time. However, due to the time lag in the integration of different observation sources and databases, the number of observed T/S profiles in 2020 is relatively small compared to other years. The Danish straits has the largest

number of observed T/S profiles, with 14213 (temperature) and 14190 (salinity) profiles. In contrast, the Gulf of Riga has the lowest number of observed profiles (466 for temperature and 457 for salinity). the south Baltic has more observed T/S profiles than the north Baltic, and temperature has clearly more observed profiles than salinity for the period 1990-2000. The number of observed T/S is also inconsistent among CTD stations. More T/S profiles are observed at BY5, BY31, ANHE, and BY2, compared to at other stations. Figure 2c shows the largest number of observed profiles at BY5 (956 for

temperature and 955 for salinity) and the smallest number of observed T/S profiles at F16 (55). In addition, the number of observed profiles at both SR5 and LL07 is less than 100.





### 3.3. assessment metrics

Considering $O_i$ and $m_i$, $i = 1,2,3,...N$, as the observed and simulated time series, respectively, to assess the reanalysis results, the metrics were calculated for the simulation by the following formula:

$\quad$ BIAS $= \overline{m} - \overline{O}$ $\hfill$ (6),

$$\text{RMSD} \sqrt{\frac{1}{N}\sum_{i=1}^{N}(O_i - m_i)^2} \hfill (7),$$

$$\text{NRMSD} = \frac{1}{\sigma_o}\sqrt{\frac{1}{N}\sum_{i=1}^{N}(O_i - m_i)^2} \hfill (8),$$

$$\text{NCRMSD} = \frac{1}{\sigma_o}\sqrt{\frac{1}{N}\sum_{i=1}^{N}\left[(O - \overline{O}) - (m - \overline{m})\right]^2} \hfill (9),$$

where overline in superscript represents the temporal average value of observed (O) or modeled ($m$) fields. SD ($\sigma$) and

$\quad$ correlation coefficient (R) are given by

$$\sigma_s = \sqrt{\frac{1}{N}\sum_{i=1}^{N}(s - \overline{s})^2} \hfill (10),$$

$$R = \frac{1}{\sigma_o \sigma_m}\frac{1}{N}\sum_{i=1}^{N}(m_i - \overline{m})(O_i - \overline{O}) \hfill (11),$$

where s is the observed (O) or modeled ($m$) fields.

In addition, a cost function (CF) following Eilola et al. (2011) was considered as a statistical measure for the simulation

$\quad$ quality. This CF was computed for each selected CTD station:

$$C = \frac{|\overline{m} - \overline{O}|}{\sigma_o} \hfill (12).$$

The CF values were calculated for each defined depth and assess the correspondence between simulations and observations as follows: $0 \le C < 1$ represents good quality, $1 \le C < 2$ represents reasonable quality, and all $C \ge 2$ signify poor agreement. This means that the results are interpreted as good if the long-term mean does not deviate from the observed mean by more

$\quad$ than plus or minus one standard deviation of the observations.

### 4. results

Based on the above reanalysis configuration, a high-resolution simulation of the Baltic Sea from 1990 to 2020 was carried out, assimilating SST satellite observations from both OSISAF and IceMap and T/S profile observations from both SHARK and ICES to constrain the model bias. This simulation was then evaluated with the observations from different datasets.

$\quad$ Finally, this simulation was used to detect the long-term and seasonal variability in T/S of the Baltic Sea.





### 4.1 Validation of the simulation results

#### 4.1.1   Comparison with in situ temperature and salinity data

NEMO–Nordic has been widely used as a model for both operational forecasting and multiscale research in the Baltic Sea (Liu and Fu, 2018; Hordoir et al.,2015; 2019; Kärnä et al, 2021). It has been carefully validated using various observational datasets, but there are still significant simulation biases against observed values (Hordoir et al.,2019; Kärnä et al, 2021). In this study, DA was used to constrain the model drift and reduce the model bias by merging observational information.   To assess the accuracy of the reanalysis simulation in capturing the temporal evolution of T/S at different depths, Figure 3 shows the evaluation values of selected CTD stations (Fig. 1) from the Baltic sub-basins, according to the CF defined by equation 11. The CF values for the temperatures are generally less than 1.0, indicating a good quality, except for the Bothnian Sea (C3, MS4, and SR5) and the bottom of south Baltic Proper (BCSIII-10). Reanalyzed temperatures are reasonable in south Bothnian Sea (SR5) and at the bottom of south Baltic Proper (BCSIII-10). In the relatively deep waters of the Bothnian Sea (C3), the temperatures have good quality, except for the temperature at 20 m. In the interior shallow waters of the Bothnian Sea (MS4), the temperature at depths of 0–10 m was poorly simulated with a CF value above 2.0, while temperatures at other depths were reasonably simulated with a CF value below 2.0. Figure 3 shows that the salinity has lower quality than the temperature, particularly in the Bothnian Sea. Compared to the north Baltic Sea, relatively high-quality salinity simulation occurred in other sub-basins.  In the Bothnian Bay, the salinity is good except for the north waters shallower than 15 m (F3) where the salinity is also reasonable simulated. In the Bothnian Sea, the salinity was reasonably reanalyzed, except for the salinity in the interior shallow waters at depths of 0–10 m (MS4) and the south bottom waters (SR5). These poor-quality salinities suggest that the DA simulation did not effectively reduce the salinity bias in the central-south Bothnian Sea. Indeed, the DA behavior among these stations is related to the number of observations available at these stations. For example, the sparse T/S observations at SR5 (Fig.2) make the correction via DA difficult, which also affects the reanalysis results in the Bothnian Sea. The CF value of salinity is between 1.0 and 2.0 at the bottom of the southern Baltic Proper (BCSIII-10), while the bottom salinity in the center of the Baltic Proper (BY15) was reasonably simulated, according to its CF value. It is noted that the number and temporal distribution of observed profiles at these stations varies, which may affect the calculation of the evaluation value. The results in Fig. 3 indicate possible areas for future improvements: large errors in surface temperatures over the Bothnian Sea: a better DA is needed in the ice edge zone; large CFs at the thermoclimatic/halocline depths also imply a need for a better DA.

To show the stability of DA's ability to constrain model bias, the long-term average seasonal cycles of T/S of the assimilated and non-assimilated simulations in the Bornholm Basin (BY5) and the east Gotland Basin (BY15) are compared in Fig.4. The salinity shows a stronger stratification at the eastern Gotland Basin (BY15) compared to the Bornholm Basin (BY5). Salinities observed in the eastern Gotland Basin also show a freshening of the upper mixed layer salinity from June to July that persists into the Autumn, but no significant seasonal variation is visible in the Bornholm Basin. Compare to observations, the simulation without DA gives saltier water above the halocline (e.g., about 1.0 PSU at BY15) and more





brackish under the halocline (e.g. about 2.0 PSU at BY5). The reason may be that model resolution cannot well resolve

bathymetry in the Danish Straits and the vertical displacement of the halocline causing too much vertical mixing in the model. As shown in Fig. 3, error of DA simulation is very small. At B5, the temperature error varies from 14 of the SD of the observations in the surface and 22% in the bottom layer. For BY15, this is 6% and 8%. For salinity the error is 23-51% of the SD of the observations at the surface and 34-38% at the bottom. In waters deeper than the halocline, the salinity bias was reduced by 2 PSU at BY5 and 0.5 PSU at BY15. In the mixed layer, the salinity errors were reduced by 1.0 PSU at BY5

and 0.8 PSU at BY15. Additionally, the temperature at these two stations shows an increasing temperature and thermal stratification in April to May and the increased vertical mixing in the autumn and winter destroys the thermocline. Model has captured these cooling/warming variabilities, but at smaller magnitudes, weaker stratification, and stronger mixing. The errors of the model temperature occurred in all vertical layers and seasons at both stations compared to the observations. After DA, these biases were generally reduced at both stations, resulting in an average decrease of 1.0 °C. Specifically, the

maximal warmer bias was reduced from 2.8 °C to 1.3 °C at 40 m at BY5 in September and from 3.2 °C to 0.5 °C at 20 m at BY15 in August. In waters deeper than the thermocline, the warmer bias at BY5 and BY15 were reduced by 0.36 °C by DA. These improvements by DA result in T/S that are in good agreement with the observations, indicating that the DA used in this study could produce an accurate T/S simulation in the Baltic Sea.

Figure 5 illustrates the vertical mean RMSD and bias between the reanalysis T/S and the ICES observations. The

analysis indicates a general overestimation of temperatures across most regions of the Baltic Sea, with notable underestimations in specific areas such as the eastern Gulf of Finland, the central Gotland Basin, the eastern coast of the Bothnian Sea, and the Bothnian Bay. The warmest bias of 0.8 °C occurred along the Swedish west coast, whereas cold biases exceeding 0.8 °C were found along the west coast of the Bothnian Sea and the eastern coast of the Bothnian Bay. Overall, the reanalyzed temperature error in the Baltic Sea was found to be less than 0.6 °C; However, larger errors occurred in the

eastern Gulf of Finland, the west coast of the Bothnian Sea, and the east coast of the Bothnian Bay. Additionally, the Skagerrak and the Danish Straits showed relatively large reanalysis errors in temperature compared to other sub-basins, which exceed 0.9 °C, with the largest error exceeding 1.3 °C along the west coast of the Bothnian Sea. For salinity, the reanalysis was dominated by negative biases, with the largest low-salinity bias in the Gulf of Riga, which is lower than -0.7 PSU. Conversely, salty biases were primarily found along the north coast of the Bothnian Bay, the west coast of the

Bothnian Sea, the Skagerrak, and the Polish coast. The simulated salinity errors are broadly above 0.5 PSU, while errors below 0.4 PSU were found in the Bothnian Sea, the Bothnian Bay, the northern Baltic Proper, and the connection region between the North Sea and the Baltic Sea. Salinity errors above 0.9 PSU were noted along the coast of the Bothnian Bay, the eastern Gulf of Finland, the Gulf of Riga, the Polish coast, and the northern Bornholm Basin. Particularly, the Gulf of Riga exhibited the largest salinity errors due to the relatively small number of observations.



**4.1.2 Sea level anomaly**

Although sea surface height (SSH) may not be affected much by T/S assimilation, especially in coastal ocean, an accurate simulation of sea surface height (SSH) can be a good indicator for reproducing the vertical dynamics of seawater. SSH observations from 26 tide gauge stations were selected to verify the reanalyzed SSH, as shown in Fig. 1. These tidal gauges are more widely distributed along the west coast than on the east coast in the Baltic Sea. A model-equivalent sea-

level anomaly (SLA) was derived by subtracting the mean sea surface height (MSSH) from the model SSH, i.e., SLA=SSH-MSSH. The MSSH depends on the time period chosen for the MSSH calculation. To avoid possible different calculation criteria between reanalyzed and observed MSSHs, SLA is used instead of SSH to evaluate the simulation performance. As the model bathymetry cannot resolve some of these tide gauge locations, the model grid points closest to the tide gauges were selected to provide an approximation of the reanalyzed SLA at the tide gauges. The SD, NCRMSD, and correlation

coefficients of the reanalyzed SLA were then evaluated against the observations, as shown in Fig. 6. The results indicated that the reanalyzed SLA variability was generally well-matched to the observations, with their SDs slightly larger than the observed values and NCRMSDs mostly smaller than 0.40. However, NCRMSDs of the reanalyzed SLA above 0.40 occurred at the stations in the Baltic-North Sea transition region (Kungsvik, Viken, Ballen, and Grena) and in the elongated Baltic sub-basins (Daugavgriva, Tallinn, and Kalix). The largest NCRMSD of SLA was found at Grena, with a value of 0.62. The

correlation coefficients between reanalyzed and the observed SLAs were above 0.80, and higher than 0.96 at LandsortNorra, Rauma, Helsinki, Rauma, Visby, Degerby, Pietarsaari, and Kemi. Relatively low correlation (smaller than 0.88) are found at Viken, Tallinn, Grena, and Ballen, with NCRMSDs above 0.50. At Rauma, the SLAs had the largest correlation (0.97) with the observations and the smallest NCRMSD (0.28), indicating the best SLA simulation. In the Gulf of Finland, the SLA was better reanalyzed on the northern coast than on the southern coast, as shown by the comparison between the Helsinki and

Tallinn stations. Overall, the simulation was found to accurately present the SSH variability in the Baltic Sea.

**4.1.3 Sea surface temperature**

The CMEMS_L4 product was used to evaluate the simulated SST. The NRMSD and BIAS between the simulated SST and the observations are shown in Fig. 7 for the simulation period.

Overall, the SST reanalysis in the Baltic Sea was good compared to the CMEMS_L4 SST, although there were some

regional differences in the simulation accuracy. Generally, the SST was warmer in the southern areas and cooler in the northern regions compared to the satellite SST (Fig.7a). Notably, SSTs were clearly overestimated along the east coast of the Baltic Proper, the south coast of the Gulf of Finland and the Gulf of Riga, Danish Straits, and Skagerrak, with the highest warm bias reaching 0.5 °C on the south coast of the Baltic Proper. Conversely, SSTs in the northern Baltic Sea were considerable underestimated, with the coldest bias reaching 1.0 °C in the northern coastal waters of the Bothnian Bay. The

NRMSD of the simulated SST against the CMEMS_L4 SST showed a slight decline from north to south and from coastal to central regions. SST errors in the Baltic Sea were typically small, with a reanalyzed NRMSD below 0.18, except in the



Bothnian Bay, southern Skagerrak, and the Neva estuary. The northernmost coast of the Bothnian Bay had the biggest reanalyzed error in SSTs with a NRMSD of 0.27, while the Neva River estuary showed a significant error with an NRMSD of 0.22. The smallest SST error, with an NRMSD value of 0.12, was found in the interior region of the Baltic Proper. In the

Skagerrak, however, the simulation showed a warm bias with an NRMSD higher than 0.19.

### 4.1.4 Mixed layer depth

The mixed layer depth (MLD) is a key indicator of the vertical dynamics of seawater characterized by nearly uniform salinity, temperature, and density. The temporal variability of the MLD is closely related to various processes occurring within the mixed layer (McCreary et al., 2001). In this study, the simulation was compared with MLDs derived from

SHARK observations (Fig. 8), using a seawater density criterion to define the MLD as the depth at which the seawater density deviates by 0.03 kg m-3 from the surface value (Chrysagi et al., 2021). Figure 8 shows that the observed MLD in the Baltic Sea shows significant spatio-temporal variations. In spring, the MLD variability in the Baltic Sea is observed to be roughly consistent with the changes in water depth. For instance, the MLD varies from 20.3 m in the Arkona Basin to 53.8 m in the eastern Gotland Basin. The Baltic MLD becomes relatively shallow in summer because of strong surface heating,

promoting the stratification and shoaling the MLD. In autumn, the Baltic MLD deepened (Fig. 8 lower left), likely due to stronger wind-induced mixing in the upper layer. In winter, the MLD reaches its deepest point of the year due to weakened solar radiation and strong wind forcing. The MLD was shallower in the far ends of the elongated Baltic Sea, including the Bothnian Bay, the Arkona Basin, and the Gulf of Finland, compared to the MLD in the central Baltic Sea. It is worth noting that the seawater in the Gotland Basin is strongly stratified, while in shallow sub-basins such as the Akona Basin, and the

Kattegat, the MLD spans almost the entire water depth due to well-mixed seawater. Compared to the observations, the MLDs derived from the reanalysis are generally smaller in the southern Baltic Sea in both summer and autumn, as well as in the southern Baltic Proper in spring. By contrast, the simulated MLD is usually larger over the entire Baltic Sea in winter, in the northern Baltic Sea in autumn, and in the north-central Baltic Proper in spring. The bias of the simulated MLD is typically less than 8.0 m. This suggests that the simulated MLDs reasonably reproduced the observed vertical distribution of

the water column across all seasons. These MLDs provide confidence in a reliable simulation of the Baltic water mass.

### 4.2 Trends and variability of temperature and salinity

Figure 9 and Table 1 provide an overview of the trends and variability in reanalyzed temperature and salinity at different depths of the Baltic Sea from 1990 to 2020. Overall, the Baltic temperature experienced a warming trend from 1990 to 2020, with an increase rate of 0.036 ºC/year at the surface, 0.023 ºC/year at 60 m, 0.041 ºC/year at 100 m, and 0.038

ºC/year at the bottom. The annual mean temperature varied between 7.06–9.72 ºC at the surface, 6.99–8.04 ºC at 60 m, 8.10–9.63 ºC at 100 m, and 4.98–6.98 ºC at the bottom (Fig. 9). At the surface, the annual maximum temperature ranged from 17.11 to 23.04 ºC. Compared to surface water, the subsurface water exhibited a narrower variability in annual maximum temperature, fluctuating between 2.05 and 2.41 ºC. The coldest water ranged from -0.23 to 5.74 ºC at these four depths



(Table 1). Fluctuations of the annual coldest water ranged from 1.60 to 2.41 ºC, showing the largest fluctuation on the
surface and the smallest fluctuation at 100 m. This indicates that the temperature variability is larger at the surface than at
other depths. The large variations in the Baltic Sea SST are caused by large changes in surface atmospheric forcing, which
directly affects the SST (Meier et al. 2019; Kniebusch et al, 2019). In contrast to the SST, the variability in the subsurface
temperature of the Baltic Sea is primarily driven by inflowing water from the North Sea and mixing with upper/lower layers.
For the salinity, the annual mean values varied from 7.85 to 8.60 PSU at the surface and from 10.11 and 10.78 PSU at the
bottom. The SDs associated with these values were 0.67 PSU for the surface and 0.54 PSU for the bottom. In intermedia
water, the salinity SD is relatively smaller, with a value of 0.35 PSU at 60 m and 0.14 PSU at 100 m.  The surface salinity
shows a freshening trend of -0.0036 PSU/year. However, the subsurface waters of the Baltic Sea became more saline at 60
m, 100 m, and at the bottom, with trends of 0.040 PSU/year and 0.049 PSU/year, and 0.012 PSU/year, respectively. It is also
noted that both the annual saltiest and freshest waters occurred at the bottom. The reason may be due to the well mixing of
the saltiest water at the bottom and the freshest water in the shallow sub-basins such as the Danish Straits.  This study
demonstrated that the SST and SSS of the Baltic Sea decreased between 1990 and 1995, but increased between 2010 and
2015. This pattern indicates a clear temporal variation of the T/S trends in the Baltic Sea.

### 4.2.1 temperature and salinity trends at the sub-basins

Figure 10 shows the spatial variability of T/S trends on the sub-basin scale. In the past 30 years, the warming trend
of the Baltic SST generally decreased from northeast to southwest, with the most pronounced increase in the Gulf of Finland
(0.07 ºC/year). The SST warming trend decreased from east to west in the northern Baltic Sea. At depths of 60 m and 100 m,
the temperature of the Baltic Sea warmed with a decreased amplitude from south to north. Temperature trends in the Baltic
Proper were smaller at 60 m compared to 100 m, reaching a high value in the eastern Gotland Basin (0.06 ºC/year and 0.07
ºC/year, respectively). Meanwhile, the Bothnian Bay displayed the smallest temperature trends at both depths, with values of
0.02 °C/year at 60 m and 0.01 °C/year at 100 m. Sea bottom temperature (SBT) trends in the Baltic Sea increased roughly
from north to south, with the large trends occurring in the Gotland Basin, the Bornholm Basin, and the eastern Gulf of
Finland, reaching 0.05 ºC/year. The SBT trend in the Bothnian Bay increased from northwest to southeast, while in the
Bothnian Sea, the trends rose from west to east, especially along the Finnish coast where the trend was the largest, reaching
0.06 ºC/year. However, the deeper waters of the Skagerrak showed a cooling trend in the SBT.

The intricate bathymetric features of the Baltic Sea restricted the water exchange between its northern and southern
regions, resulting in significantly lower salinity trends in the northern Baltic Sea compared to those in the southern Baltic
Sea. In general, the salinity trend in the northern Baltic Sea was predominantly negative during the period 1990–2020, while
a positive trend of salinity was prevalent in the majority of the Baltic Proper. In the Skagerrak, the salinity tended to decrease
from the surface to 60 m and increase from 100 m to the bottom. The SSS trend in the Baltic Proper exhibited a gradual
decrease from 0.010 PSU/year in the south to no significant change in the intermediate areas, followed by a decrease of -
0.012 PSU/year in the northern Baltic Sea, Kattegat, and Skagerrak. Particularly, a freshening trend of SSS was found on the



Polessky coast. At depths of 60 m and 100 m, the salinity became saltier in the Baltic Proper, in the Gulf of Finland, and along the coast of Skagerrak, while a slight desalination occurred in the northern part of the Baltic Sea, with no significant variability in the central part of Skagerrak. The sea bottom salinity (SBS) in the Baltic Sea demonstrated a decreasing trend
in the north regions and an increasing trend in the south regions. Specifically, the SBS trend in the Baltic Proper showed a decreasing trend from northwest to southeast. In the transition zone, the SBS tended to be saltier in the Swedish offshore area while experiencing desalination in the Danish offshore area (Fig. 9). Additionally, the SBS exhibited a salinization trend in the southern sills of the Danish Straits. Notably, the tendency for increased salinity was more pronounced at deeper waters, as evidenced in both the Baltic Proper and the Bornholm Basin.

**4.2.2 interdecadal trends in temperature and salinity**

Figure 11 reveals notable interdecadal linear trends in T/S within the Baltic Sea, characterized by significant variations across the decades. From 1990 to 2020, a prevailing warming trend in the interdecadal temperature is evident. At both the surface and bottom layers, the first decade witnessed a cooling phase, which was followed by an accelerated warming trend from 2000 to 2020. Specifically, the SST experienced a decrease of 0.96 ℃ from 1990 to 1999, then an
increase of 0.30 ℃ from 2000 to 2009, culminating in a rise of 1.17 ℃ from 2010 to 2020. Warming trends were also detected at depths of 60 m and 100 m. At 60 m, the largest warming trend occurred in the second decade at a rate of 0.043 ℃/year, while at 100 m it occurred in the first decade at a rate of 0.107 ℃/year. In contrast, the salinity decadal trends at the surface and bottom were less pronounced than those at intermediate depths. In the first decade, salinity at both the surface and 100 m showed a decreasing trend, which is opposite to the interdecadal variability of other salinities. In the second
decade, the salinity trend was 0.001 PSU/year at the surface and 0.033 PSU/year at the bottom, while the intermediate layers exhibited more substantial changes, with salinity increasing at 0.087 PSU/year at 60 m and 0.064 PSU/year at 100 m. The most significant decadal variation in salinity reached 0.166 PSU/year at 60 m between 2010 and 2020. Notably, the surface salinity trend in the second decade was the smallest at 0.001 PSU/year. At 100 m, the salinity increased from 0.007 PSU/year in the first decade to 0.108 PSU/year in the third decade. Additionally, the reanalyzed salinity revealed that the Baltic salinity
had an accelerated increase over the investigated time period at 100 m and a gradual decrease at the bottom.

**4.2.3 Seasonal variations in temperature and salinity**

Figure 12 provides an analysis of the seasonal variations of T/S within the Baltic Sea across various decades. At both the surface and the bottom, the temperatures showed an increase from March to July, followed by a decrease from August to the subsequent February. In contrast, temperatures at 60 m exhibited a distinct seasonal pattern, decreasing from January to
April and then rising from May to December. Over the past thirty years, the monthly mean temperatures at the surface and 60 m reached their minimum in February and April, with values of 1.30–1.82 ℃ and 3.00–3.91 ℃, respectively. From August to December, the monthly mean temperature decreased from 15.81–17.06 ℃ to 3.87–4.86 ℃ at the surface and increased from 3.69–4.40 ℃ to 4.84–5.52 ℃ at 60 m. However, the temperature at 100 m didn't exhibit significant seasonal



variation. A discernible warming trend in monthly temperatures has been found in the past decades. The interdecadal variation in monthly mean SST reached its maximum (1.19 ºC) in July during the previous two decades and its minimum (0.01 ºC) in August over the last two decades. The Baltic monthly mean SSS showed a visible seasonal variation, decreasing from January to May and subsequently increasing from May to December. Furthermore, significant differences in the monthly mean salinities were noted across the three decades. For example, the difference in monthly SSS between the first- and second-decade was 0.307 PSU in March, while it was less than 0.018 PSU in both August and September. There was no 410 evident seasonal variation in the salinity of the subsurface water. The variation in monthly mean salinity was highest between the first and third decade in March (0.791 PSU at a depth of 100 m) and lowest in September between the second and third decade (0.0079 PSU at the surface).

## 5. Discussions

In this study, the quality of the reanalysis is affected by errors related to forcing, model physics, model spatial resolution, 415 initial conditions, data assimilation scheme and availability of observations. The first four factors are common to both DA and non-DA simulations while the last two factors only relate to the DA simulation. The uncertainties in the forcing include those from UERRA, especially in precipitation and nearshore winds, from bulk parameterizations of the surface fluxes, river discharges and lateral boundary conditions. A preliminary tuning on the precipitation has been made, leading to its reduction by 20%. More optimization on the forcing can be made to reduce bias in the non-DA simulation in the future. Below we 420 further discuss potential error sources in the reanalysis and also intercompare our results with other existing studies.

### 5.1 Reanalysis bias

As shown in Kärnä et al. (2021), the NEMO-Nordic model equations are imperfect, leading to insufficient details in some processes. Significant warm bias in the open Baltic Sea in the non-DA simulation may be caused by overestimated vertical mixing. Further, the 2 nautical-mile horizontal resolution used in this study cannot resolve the complicated Baltic 425 coastline and channels, especially in Baltic-North Sea transition region (She et al., 2007b), resulting in inaccurate sea level and Baltic-North Sea water exchange. Higher resolution should be applied to resolve the shallow coastal regions and areas with narrow water passages. On data assimilation scheme, please write

In addition to the uncertainties in model physics, numerics and forcing, the large biases in the reanalysis simulation are mainly caused by imperfect initial conditions and high uncertainties of coastal observations (Hordoir et al, 2019; Kärnä 430 et al. 2021). The model underestimated the salinity and overestimated the temperature in the deeper waters of the Bornholm Basin, while the DA simulation improved the model results (Fig. 3). The MLD biases between model results and observations can influence bottom water ventilation, especially in shallow waters (Mohrholz, 2018, Liu et al., 2014). However, DA significantly reduced the reanalysis bias at the beginning of the simulation and remained relatively small during integration compared to the model alone (Liu et al. 2013, 2017). The quality of deep-water salinity reanalysis is poor



at some stations. For example, in the southern Bothnian Sea (SR5), the CF for reanalyzed bottom salinity showed a value of 2.62 (Fig. 4). The reasons for this may be that deep water salinity varies very little and lack of observations at these stations and the adjustment of LSEIK is directly related to the spread in the sample ensemble. Data assimilation scheme is another source of uncertainty for the reanalysis. This study used a constant inflation factor to adjust the ensemble variance for this long-term simulation, which may not be appropriate for a specific period or position. For example, the constant inflation

factor in this study was successful in constraining NEMO-Nordic in the east Gotland Basin (BY15), but not at the southern Bothnian Sea bottom (SR5). Therefore, an inhomogeneous inflation factor is needed for a high-quality salinity simulation when using the DA method in the Baltic Sea.  For example, a varied inflation factor can be used for different locations or sub-basins.

    Furthermore, it should be noted that the reanalysis assimilated the OSISAF SST observations in this study, while the

validation was performed using a satellite SST product from CMEMS. Liu and Fu (2018) has demonstrated that the DA configuration of this reanalysis could effectively assimilate SST observations over the Baltic Sea. However, the differences between the DA and validation satellite SSTs could potentially impact the accuracy of the SST validation process. It is worth noting that the SST discrepancies between OSISAF and CMEMS_L4 varied in time and space (not shown). For instance, compared to the CMEMS_L4 SST, the OSISAF SST was warmer in both the Gulf of Bothnia and the Bornholm Basin in

2005, but colder in 2006. Similarly, the OSISAF SST in 2006 was warmer in the Gulf of Finland and colder in the central of the Baltic Proper. When the reanalyzed SST are compared with those of CMEMS_L4, our results found a more pronounced cold bias in the Gulf of Bothnia. This indicates the cumulative impact of the DA adjustment on the generated SST. These results highlight the importance of considering such discrepancies for a comprehensive assessment of the reanalysis results and their implications for understanding SST variations.

**5.2 Observational limitations in the Baltic Sea**

    This study utilized observations from both in-situ and remote sensing sources. However, it is important to note that monitoring systems in the Baltic Sea only observe the surface waters or the deep waters of the Baltic Sea at predefined standard stations and depths (She et al, 2020), and the number of T/S profile observations is inhomogeneous in Baltic sub-basins (Fig.2). This may lead to relatively large uncertainties of T/S in the reanalysis in areas with few observations.

Similarly, the accuracy of assimilated satellite SSTs in coastal waters can be affected by high uncertainties due to complicated retrieved conditions and radio frequency interference by land (Teruzzi et al. 2014, Liu and Fu, 2018). The constant observation error used for the entire Baltic waters during the DA process of SST also suggests that the satellite observations in the coastal waters of the Baltic Sea need to be applied with caution during SST assimilation. For example, larger observation errors are applied in the coastal region or coastal observations are discarded.

There is significant variation in the number of T/S profile observations at the selected stations in Fig. 3. The confidence of the evaluation function in Eq. 12 depends on both the number of observations used for evaluation and their temporal distribution. For example, there are only 34 T/S profile observations at F16, and these profiles are discretely distributed over



the simulation period, almost one per year. Therefore, due to the discrete distribution of its observations, the low evaluation value of F16 hardly gives us confidence in its high-quality simulation. On the other hand, the BY5 observations are available

for nearly all simulation months, which ensures a high reliability of the validation. Another example is that BY15 has a higher confidence in the evaluation than C3 because the profile observations for C3 only cover years after 1999, but those for BY15 cover all years of the simulation.

Additionally, the integration of observational information can significantly enhance the simulation accuracy by DA and provide a more precise and reliable distribution of T/S which well match with observed values, as depicted in Fig. 4. In this

study, the DA method spreads T/S increments around the observation locations according to a predefined influence radius. This approach reduces the impact of DA on altering the dynamics of the model, thereby preserving the integrity of the simulation. However, it is important to note that this research is constrained by the limited spatial and temporal coverage of the observational data used in DA. This scarcity can potentially affect the quality and robustness of the DA simulation, as smaller or unevenly distributed datasets might limit the model's ability to accurate representation of T/S dynamics across the

studied region, particularly in areas where observational data is sparse or absent. Consequently, the overall effectiveness of DA could be influenced, potentially distributing the model's dynamic balance in various regions and causing inaccuracies in trend distributions between and within sub-basins. The implications of these alterations in model's dynamics on trend analysis are not yet fully understood. Further investigation is required to determine the extent to which these model adjustments influence the overall trend patterns and their accuracy within the specific sub-basin contexts.

**5.3 Comparison with other data assimilation studies of the Baltic Sea**

Several DA studies have focused on the historical reanalysis of T/S in the Baltic Sea (e.g., Fu et al., 2012; Liu et al., 2013; Axell and Liu, 2016). DA methods such as Three-Dimensional Variational (3DVAR) (Fu et al., 2012), Ensemble 3DVAR (Axell and Liu, 2016), Ensemble Optimal Interpolation (EnOI) (Liu et al., 2013), and Local Error Subspace Transform Kalman Filter (LESTKF) (Baltic Sea Physics Reanalysis, 2023) were used in reanalyzing historical T/S record

with assimilation of the T/S profiles. The observation error covariance matrix considered as a constant for T/S throughout the water column in previous studies (Fu et al., 2013; Liu et al., 2013; Axell and Liu, 2016). In the present study, LSEIK was used in a 31-year reanalysis of T/S in the Baltic Sea with a circulation model, while the observation error varied with increasing water depth for a profile, which is helpful to constrain the bottom simulation with an ensemble DA method like LSEIK. This study showed a slightly better bottom salinity compared to Axell and Liu (2016) in the Baltic Proper (BY15)

and the Bornholm Basin (BY5). For example, the bottom salinity of Axell and Liu (2016) held a lower salinity bias of 0.23 PSU at BY5 and 0.10 PSU at BY15. The bottom temperatures from Liu et al. (2013) were also 0.5 ºC and 0.3 ºC higher than observed values at BY15 and BY5, respectively. However, the bottom T/S provided by Baltic Sea Physics Reanalysis (2023), this study, and Fu et al. (2012) is in good agreement with the observations at both BY5 and BY15. It is also noted that there was an overestimated local variability in the bottom salinity in the Baltic Sea Physics Reanalysis (2023), which

caused a sudden decrease of 1.9 PSU after the inflow in 1996 and an increase of 0.7 PSU in the spring of 2009 in the bottom



salinity of the Baltic Proper (https://catalogue.marine.copernicus.eu/documents/QUID/CMEMS-BAL-QUID-003-011.pdf), these are not recorded in both the ICES and this study. It was also found that Fu et al. (2012) incorrectly calculated the baroclinic inflows of 2006 (Lehmann et al., 2022), which was not an issue in this study (not shown). Furthermore, all reanalysis yielded a more reasonable mixed layer depth than NEMO-Nordic compared to the SHARK observations.

### 5.4 Comparison with other studies on long-term variability in the Baltic Sea

Derived from the satellite data products, the Baltic SST increased by 0.059 ºC/year for 1990−2018 (Siegel and Gerth, 2019), 0.05 ºC/year for 1982-2013 (Stramska and Bialogrodzka, 2015), 0.041 ºC/year for 1982–2012 (Høyer and Karagali, 2016), and 0.048 ºC/year for 1982-2021 (Jamali et al., 2023). Compared to these estimations from satellite products, the SST trend reported in this study is a slightly smaller at 0.036 ºC/year. By using in-situ observations from a period of 35 years, Liblik and Lips (2019) also found the SST was warmed by 0.03–0.06 ºC/year in most of the Baltic Sea during 1982−2016, with a higher rate of warming (> 0.06 ºC/year) occurred in the shallower, closed-end areas of the gulfs. Stockmayer and Lehmann (2023) detected a warming in SST of about 0.03–0.04 °C/year in the Baltic Proper for 1950-2020. Our study and Stockmayer and Lehmann (2023) have similar findings for the SST trends in the Arkona Basin, the Bornholm Basin, and the Gotland Basin. However, Stockmayer and Lehmann (2023) showed a smaller trend of SST in the Gulf of Finland. Additionally, the annual mean SST trends derived from their model were notably lower than those from observations. Additionally, the SST trends presented in this study are similar to those reported by Liblik and Lips (2019) for the Bothnian Sea, the interior of the Bothnian Bay, and the West Gotland Basin, but unlike Liblik and Lips (2019) in the Gulf of Finland and Bothnian Bay, which reported a higher warming rate for the SST from the coast to open waters. The present study also finds that the SST variability is higher in the Gulf of Finland and the northern Gotland Basin than in other Baltic sub-basins, especially in the Bornholm Basin, which is similar to findings of Stramska and Bialogrodzka (2015) and Stockmayer and Lehmann (2023), but differs from Liblik and Lips (2019). However, in the northern Baltic Sea, the SST trends derived from this study are larger than those from Stramska and Bialogrodzka (2015). For example, in the central part of both the Bothnian bay and the Bothnian Sea, this study and Stramska and Bialogrodzka (2015) presented SST trends of less than 0.05 K/year and more than 0.05 ºC/year, respectively. Moreover, this study showed an opposite SBT trend. Furthermore, this study showed SBT trends were smaller than SST trends, with a largest increase in the SBT of the Bornholm Basin. In addition, the Baltic Sea Physics Reanalysis (2023) showed the temperature trend was 0.05 ºC/year at the surface, 0.045 ºC/year at 60 m, and 0.06 K/year at both 100 m and the bottom for the period 1993−2022 (https://marine.copernicus.eu/access-data/ocean-monitoring-indicators/baltic-sea-subsurface-temperature-trend-reanalysis). Tabel 1 shows temperature trends presented in this study are evidently smaller than those from the Baltic Sea Physics Reanalysis (2023).

Regarding the salinity of the Baltic Sea, Liblik and Lips (2019) reported a freshening trend (-0.005 to -0.014 PSU/year) in the upper layers and a salinization trend (0.02–0.04 PSU/year) in the deep layers for the period 1982−2016. These findings are similar to the results of this study. Stockmayer and Lehmann (2023) also reported a freshening trend in



the surface salinity of the Baltic Proper, with values ranging from -0.005 to -0.021 PSU/year according to ICES data, and from -0.008 to -0.012 PSU/year based on model simulations for the period 1950-2020, but salinity trends in the Bornholm Basin and the Arkona Basin were opposite to those in this study and Liblik and Lips (2019). Notably, the model-derived annual mean salinity trends presented by Stockmayer and Lehmann (2023) were notably lower than those from ICES. Moreover, the Baltic Sea Physics Reanalysis (2023) reported a different trend of salinity of the Baltic Sea, showing an increase in the surface salinity of 0.012 PSU/year for the period 1993−2021. This study also found a higher salinity trend

(0.01 PSU/year) near the southern coast of the Baltic Sea compared to Liblik and Lips (2019) for the period 1990−2010. At depths of 60 m and 100 m, the Baltic Sea Physics Reanalysis also shows a smaller positive trend in salinity compared to both this study and Liblik and Lips (2019). These differences in T/S trends may be attributed to the use of different datasets with varying accuracy and horizontal resolution from models or satellite products. Furthermore, these studies cover different time periods and have uncertainties from the observation or modeling. The climate varies in time and space. For instance, over the

past century, more anticyclonic circulation and westerly winds have led to a warmer climate and less sea-ice cover (Stramska and Bialogrodzka, 2015). Therefore, it is not surprising that other studies have yielded different findings from this study. It is worth noting that this study used consistent atmospheric forcing and varying observation errors at different depths. Consequently, the results presented in this study are considerable and reasonable when compared to the analysis based on the model or observations alone. Additionally, as presented in Stockmayer and Lehmann (2023), the considerable discrepancies

between observed trends and those detected from simulations highlight the benefits of the observation-model melding for effective trend and variability analysis.

**6. Summary and conclusions**

A multi-decadal simulation was conducted using the DA method to reproduce the Baltic state from 1990 to 2020. This simulation was rigorously validated using both satellite and in-situ observations. The results showed reasonable accuracy in

key parameters such as sea level, T/S, and mixed layer depth, which made it a reliable source of information for assessing variations in the Baltic Sea in the past decades. A reasonable assessment of T/S variability in the Baltic Sea was provided over the past 30 years. The conclusions from the findings in this study include:

1.  The reanalysis provided a reasonable dataset to validate the research on the Baltic Sea.
2.  It was revealed that the temperature in the Baltic Sea has been warming at various depths over the past 30 years.

The warming rate was found to be 0.036 ºC/year at the surface, 0.023 ºC/year at 60 m, 0.041 ºC/year at 100 m, and 0.039 ºC/year at the bottom. The Baltic Sea experienced a more significant warming trend in its southern sub-basins compared to its northern sub-basins, and the water below the thermocline is warming at a faster rate than the mixed-layer water.
3.  Moreover, the study showed that the salinity in the Baltic Sea exhibited a desalination rate of -0.0036 PSU/year at

the surface from 1990 to 2020, particularly in the northern Baltic Sea. However, at depths of 60 m, 100 m, and the



bottom, the salinity displayed an increasing trend of 0.040 PSU/year, 0.049 PSU/year, and 0.012 PSU/year period, respectively. The subsurface demonstrated a higher trend in salinity compared to both the surface and bottom. Additionally, the Baltic waters shallower than 100 m displayed a larger increasing in salinity during the period 2010-2020 than the period 1990-2009.

4.  Furthermore, the seasonal variations of T/S in the Baltic Sea varied considerably at different depths. Salinity, for example, showed a clear seasonal variation at the surface but not in deeper layers, whereas the temperature did not exhibit noticeable seasonal variation at a depth of 100 m.

**Code and Data availability**

All the scripts and the post-processed data for the figures can be found here: https://doi.org/10.5281/zenodo.13961375

**Author contributions**

YL and LA conceptualized the study and YL carried out the experiment run. JS advised on the paper's structure. LA collected the observational data. YL drafted the paper with contributions from JS and LA. All authors contributed to the scientific discussion of the methods and results, reviewed and edited the final paper.

**Competing interests**

The authors declare that they have no conflict of interest.

**Acknowledgment**

This research presented in this study is supported by the Copernicus Marine Environment Monitoring Service (CMEMS).





**Table 1. The temperature and salinity variability in the Baltic Sea over the period 1990-2020**

| Parameter | Trend [year $^{-1}$] | Mean | Maximum | Minimum |
|-----------|---------------------|------|---------|---------|
| SST [ºC]  | 0.036   | 8.41  | 23.04 | -0.23 |
| T60 [ºC]  | 0.023   | 7.44  | 8.08  | 1.57  |
| T100 [ºC] | 0.041   | 8.99  | 6.64  | 3.51  |
| SBT [ºC]  | 0.038   | 5.78  | 12.51 | 1.33  |
| SSS [PSU] | -0.0036 | 8.18  | 9.94  | 5.74  |
| S60 [PSU] | 0.040   | 4.12  | 9.05  | 6.32  |
| S100 [PSU]| 0.049   | 4.79  | 9.94  | 7.48  |
| SBS [PSU] | 0.012   | 10.41 | 12.18 | 8.81  |






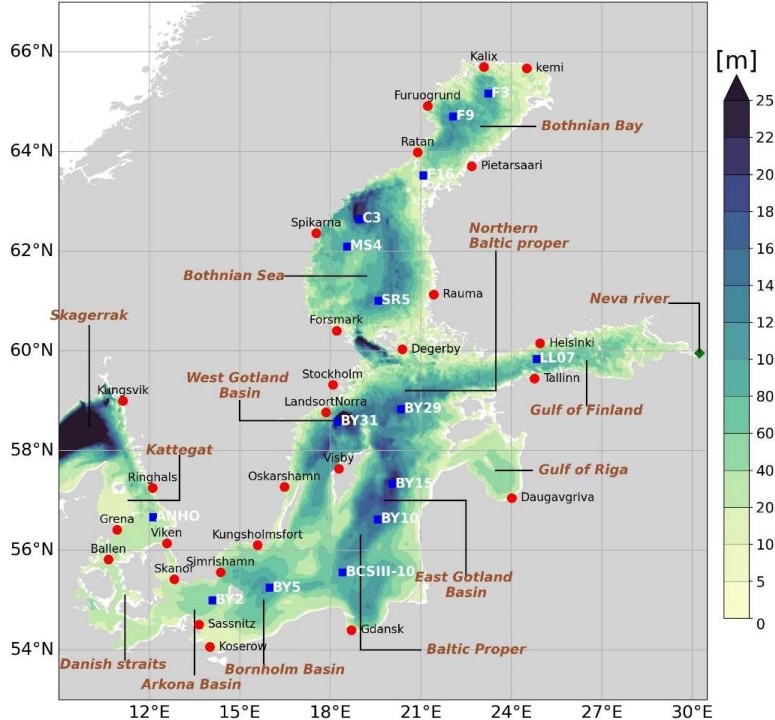

**Figure 1. Geographical domain and bathymetry of the Baltic Sea. Red dots and blue squares indicate tide gauges and station locations, respectively.**




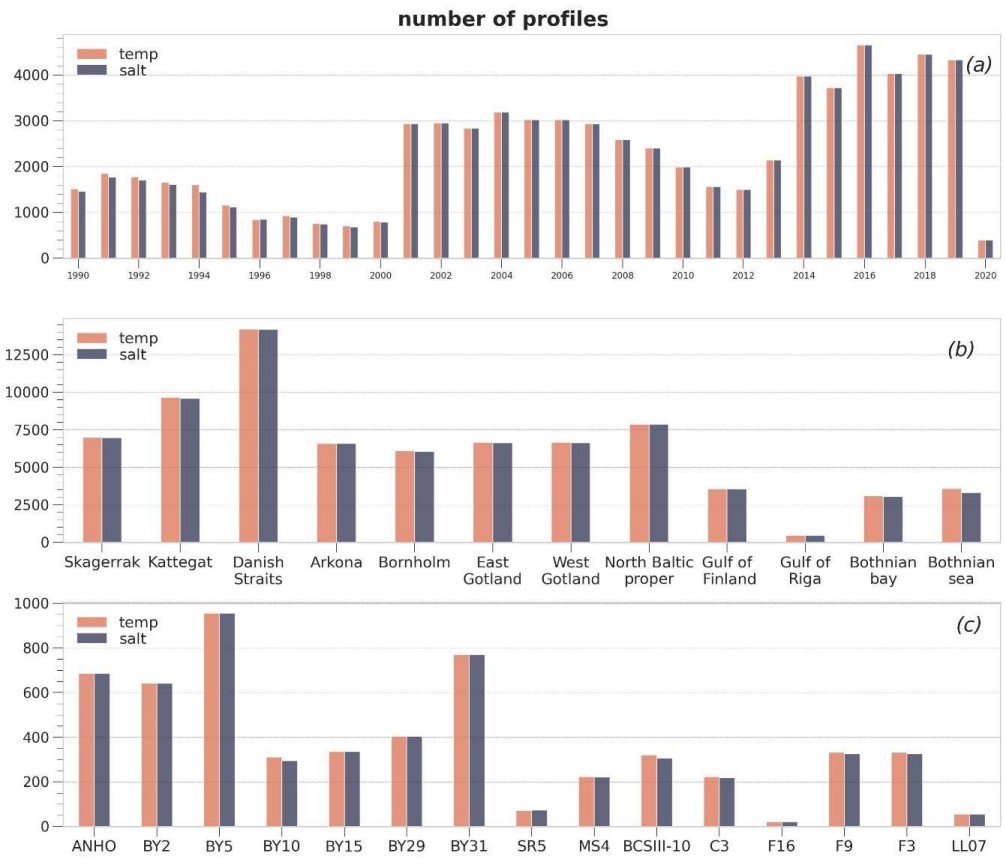


**Figure 2. Annual number of profiles (a), total number of profiles in the Baltic sub-basins (b), and number of profiles at the selected stations for the period 1990-2020 (c).**







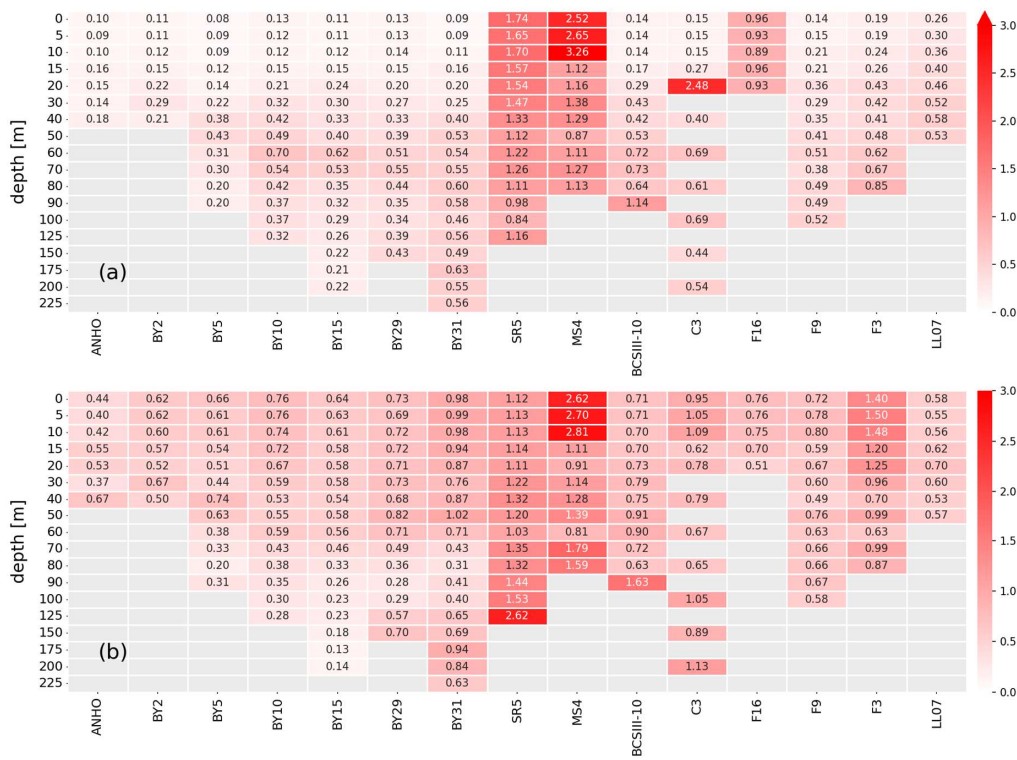

**Figure 3.** Cost function values derived from monthly time series of sea temperature (a) and salinity reanalysis (b) at the selected
monitoring stations (see Fig.1) and number of profiles at the selected monitoring stations (c) from 1990 to 2020.




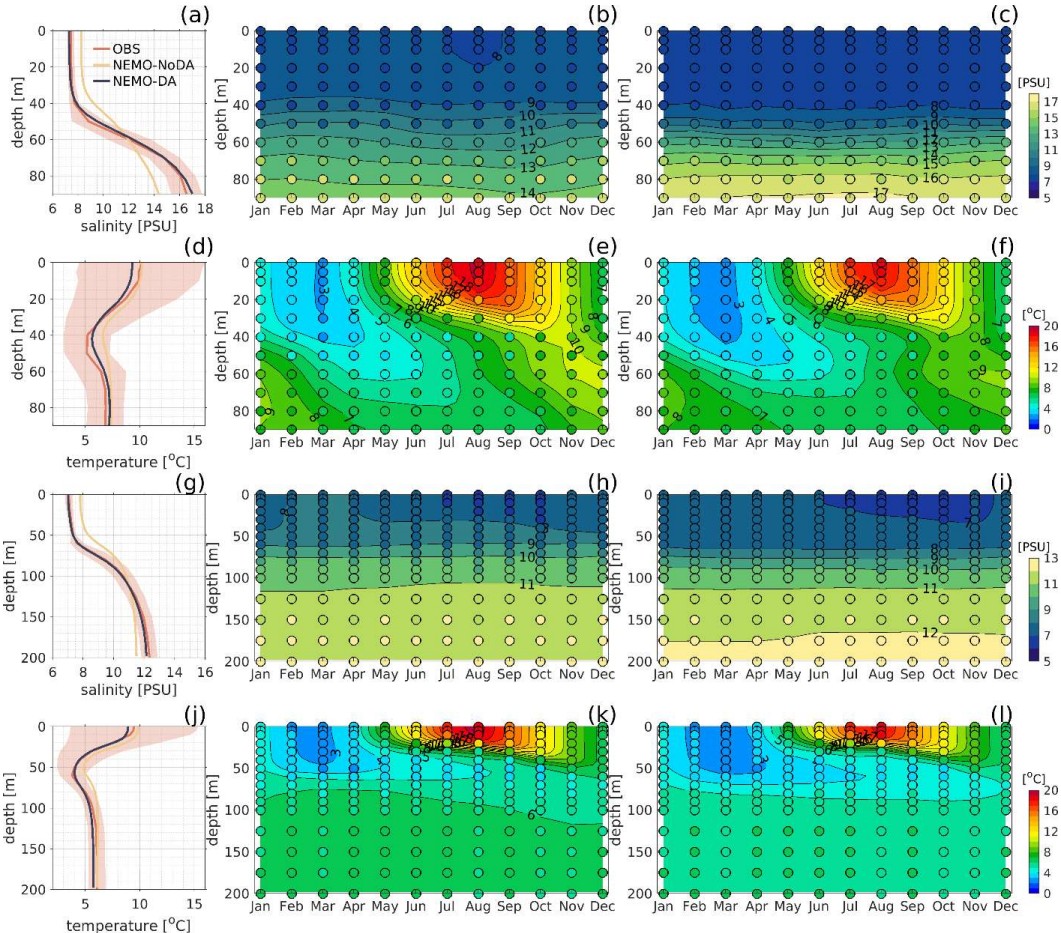

**Figure 4. Monthly, seasonal and period averages of salinity and temperature at BY5 (a-f) and BY15 (g-l) for the period 1990-2020. Time averages (a, d, g, j) are shown over the entire water column and the standard deviation of observations over the period are shown as a grey shaded area. The seasonal variable simulated from model with (c,f,i,l) and without (b,e,h,k) data assimilation are compared with the observational monthly average values (the filled circles).**





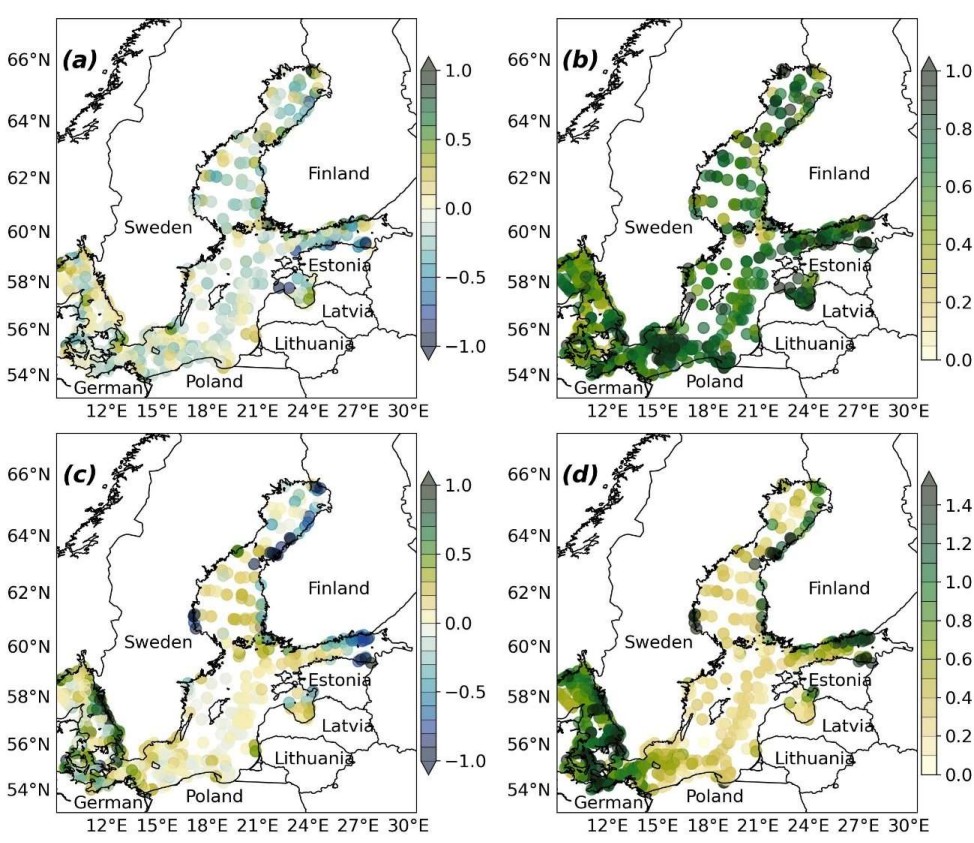

**Figure 5. Global averaged bias (a,c) and RMSD (b,d) of salinity (a,b) and temperature (c,d) from reconstructed fields relative to all**
**ICES profiles from 1990 to 2020.**





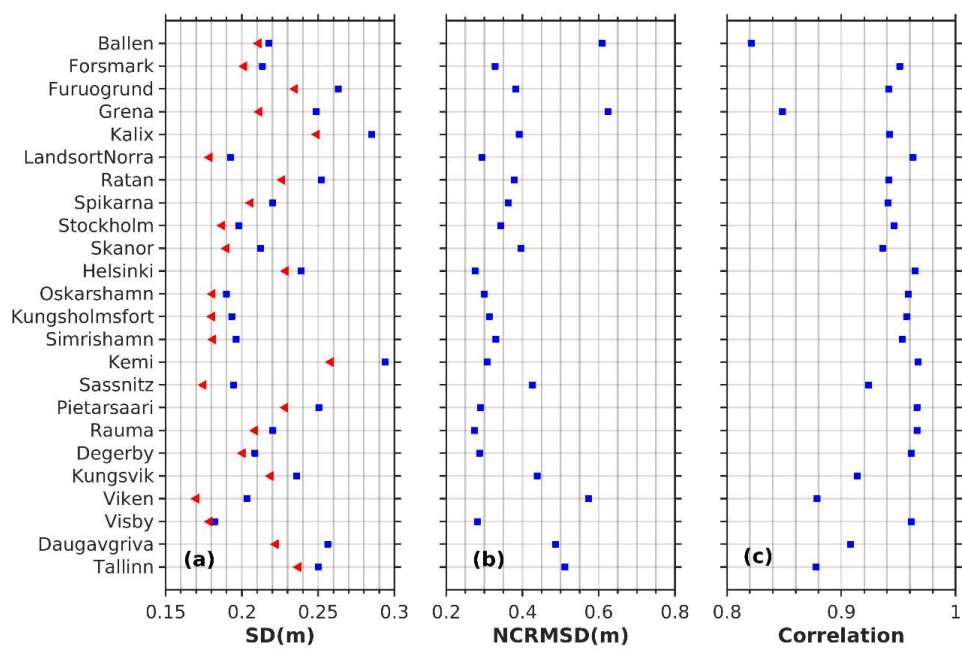

**Figure 6. Sea surface height anomaly error metrics for the period 1990-2020. Blue and red symbols denote the metrics from model and observations, respectively. See Fig. 1 for station locations.**






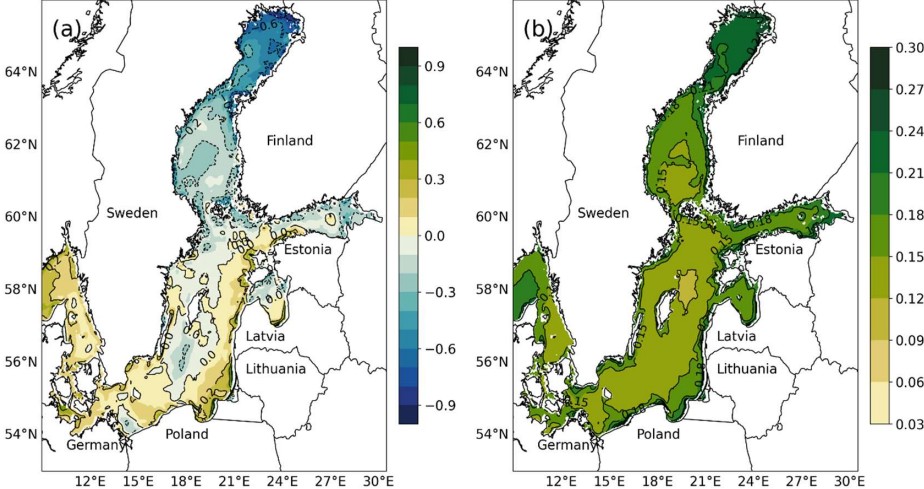

**Figure 7.** The overall averaged (a) bias, (b) NRMSD of sea surface temperature relative to the CMEMS L4 SST product for the period 1990-2020.







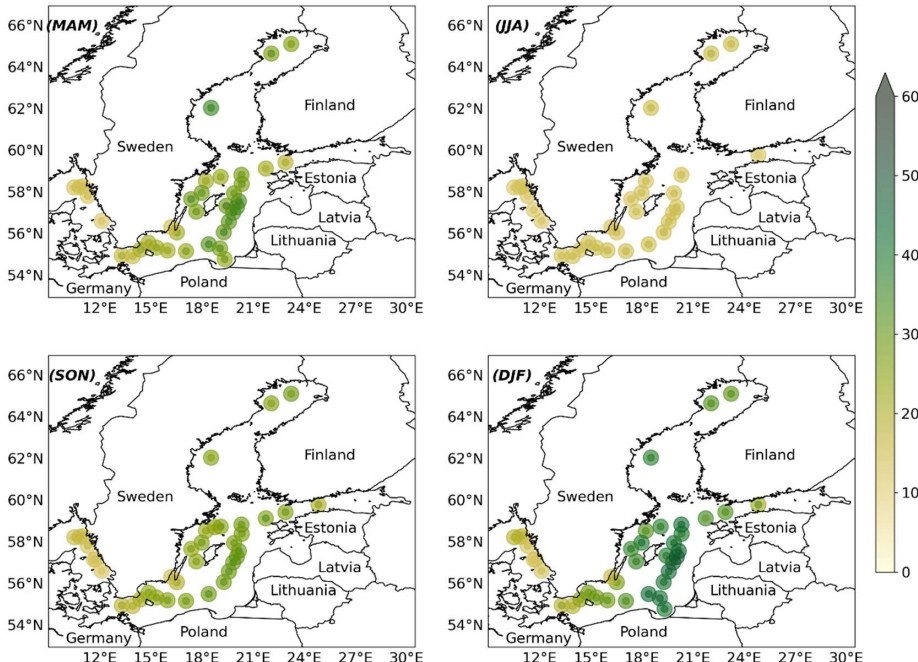

**Figure 8. Seasonal mixed layer depth for the period 1990-2020: MAM: March-April-May; JJA: June-July-August; SON: September-October-November; DJF: December-January-Feburary. The dots depict the results derived from the reanalysis (outer dots) and the observations (inner dots), respectively. The colorbar shows the size of the mixed layer depth.**



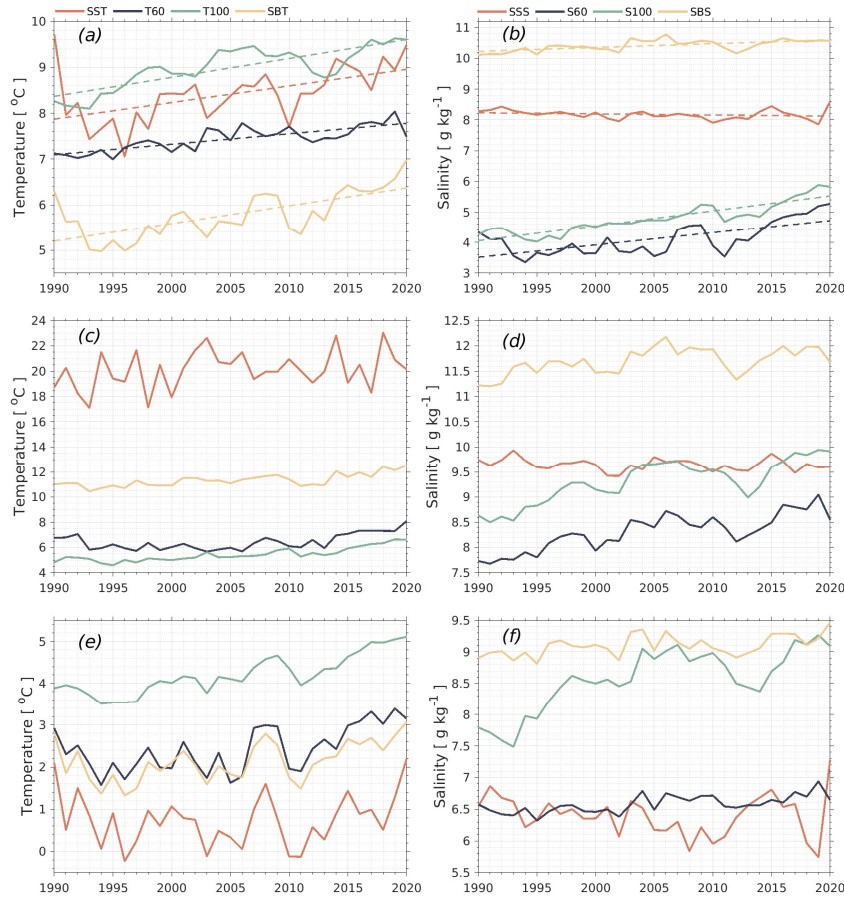

**Figure 9. The annual mean (a,b), maximal (c,d), and minimal (e,f) temperature and salinity at the surface, 60 m, 100 m, and bottom of the Baltic Sea. Linear trends are showed by dotted lines.**




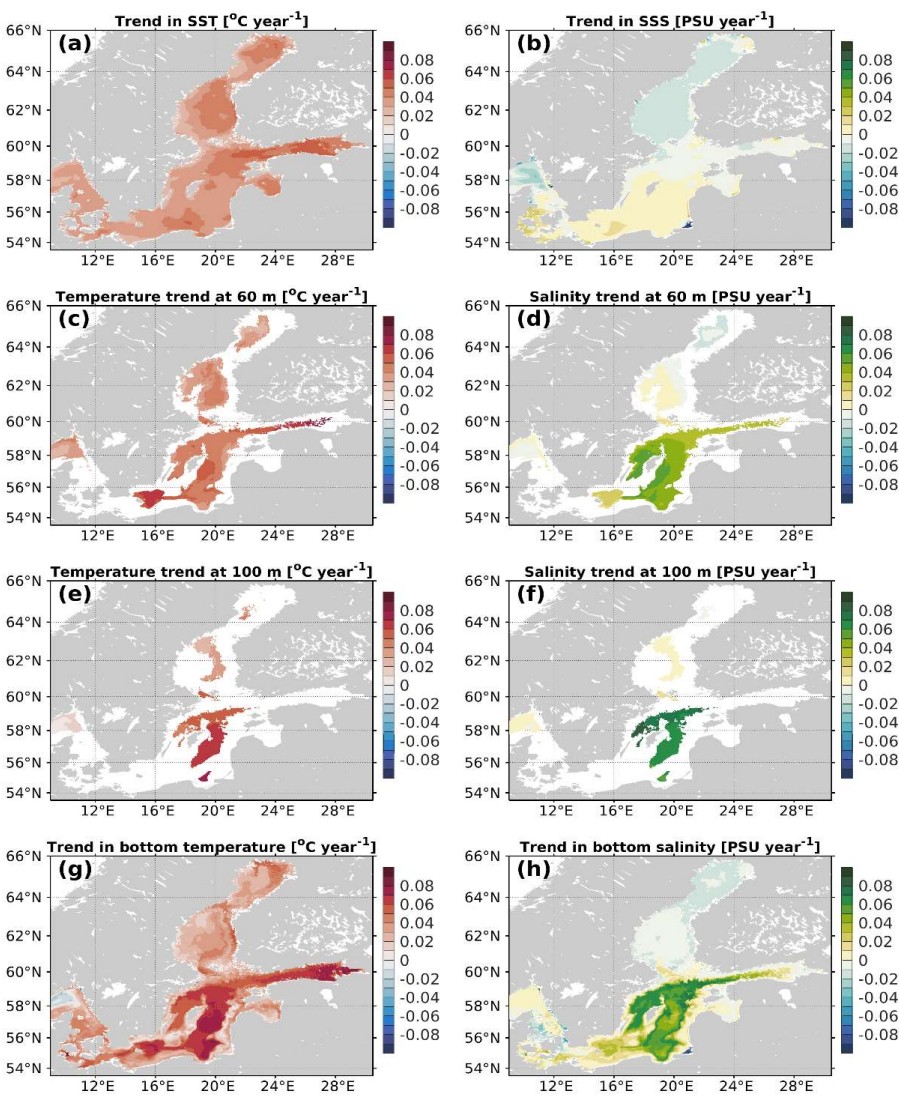

**Figure 10. Linear trends in the water temperature and salinity in the Baltic Sea during the period 1990-2020, derived from the reanalysis.**





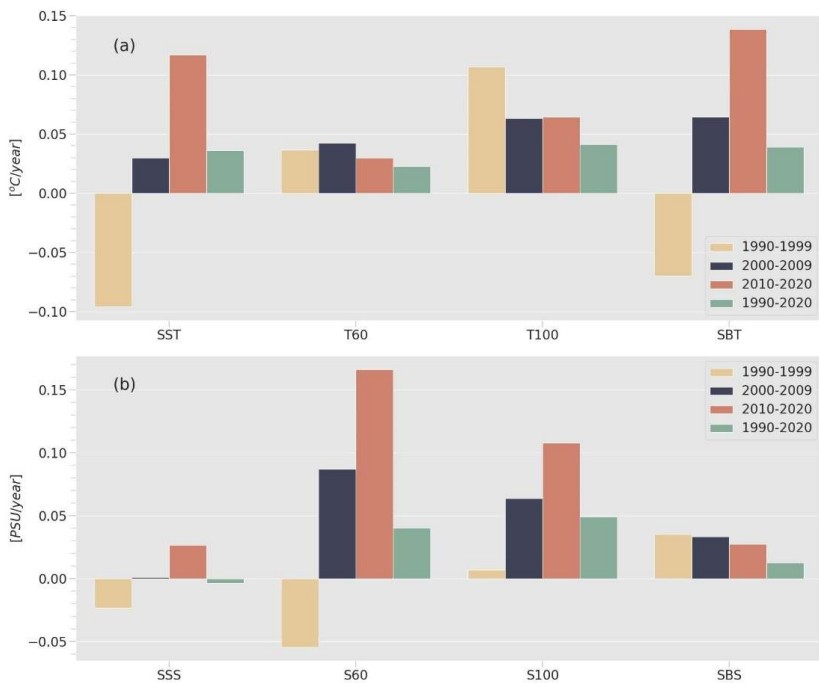

**Figure 11. Linear trends in temperature (a) and salinity (b) at different depth for the Baltic Sea over a decade.**





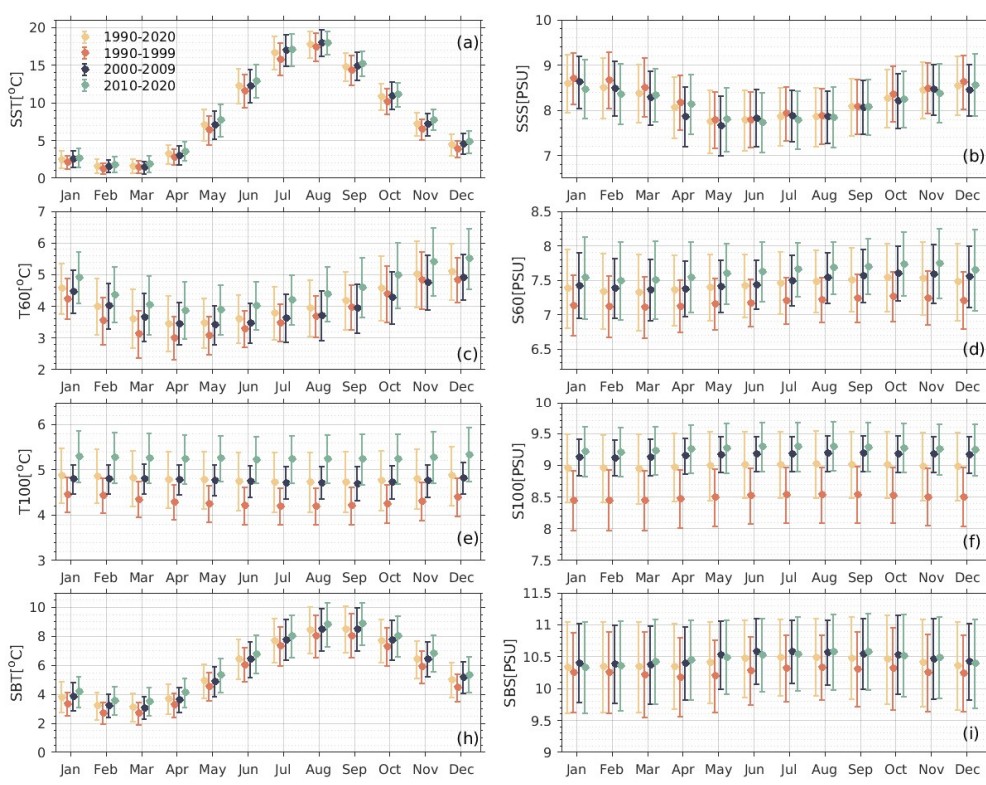

**Figure 12. The monthly temperature and salinity of the Baltic Sea at the surface, 60 m, 100 m, and the bottom for different decades, derived from reanalysis.**




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
