# Peer review of "An assessment of the variability in temperature and salinity of the Baltic Sea from a simulation with data assimilation for the period 1990 to 2020"

_EGUsphere, 2024_

## Referee Comment (RC1)

Comments on the MS by Ye Liu, Lars Axell, and Jun She
**An assessment of the variability in temperature and salinity of the Baltic Sea from a simulation with data assimilation for the period 1990 to 2020**
https://doi.org/10.5194/egusphere-2024-3283

The MS presents the application of Baltic Sea physics reanalysis system, based on the NEMO-Nordic ocean model engine set up on the 4 km (2 nautical mile) grid. The model is driven by atmospheric forcing from the UERRA reanalysis product, open boundary data from the northeast Atlantic barotropic surge model, and daily river discharge taken from a dataset by the EHYPE model. Observational data (T/S profiles, SST from remote sensing) are assimilated using the Local Singular Evolutive Interpolated Kalman (LSEIK) filter. The reanalysis system has been developed within the activities of the Baltic Monitoring and Forecasting Centre (BalMFC) of the Copernicus Marine Service (CMEMS), where the authors play an active role.

Presentation of the reanalysis system, including a validation study, is followed by an analysis of mean, variability, trends and seasonal cycle of temperature and salinity.

The MS is interesting, but before publication I recommend clarification of the following points.

**A) Relation of the present reanalysis data to the data downloadable from the Copernicus Marine Service**

Baltic Sea Physics Reanalysis data, downloadable from the CMEMS portal (https://data.marine.copernicus.eu/product/BALTICSEA_MULTIYEAR_PHY_003_011/description, https://doi.org/10.48670/moi-00013) has growing use in research, reflected in numerous scientific publications. Presently the reanalysis 1993-2023 has been upgraded to 2 km (1 nautical mile) resolution. For the users of CMEMS reanalysis data it remains unclear whether this publication presents the same or alternative data set. It should be explicitly written in the MS and reflected in the section Code and Data availability.

**In case the analyzed data set is an alternative to the CMEMS downloadable data**, the present reanalysis data originals should be made available. The MS refers only to post-processed data used for drawing the figures: https://doi.org/10.5281/zenodo.13961375. Also, comparative statistics of the present data and of https://doi.org/10.48670/moi-00013 should be presented.

**B) Observational data sets**

The MS mentions the use of T/S profile data adopted from the SHARK and ICES data bases. Other types of in-situ T/S data (ferrybox, gliders, Argo floats etc) available from https://doi.org/10.48670/moi-00032 and used for https://doi.org/10.48670/moi-00013 are not referred. If the use of observational T/S data is indeed limited, it needs to be clarified.

**C) Statistics at different depths over the whole basin**

The results contain curious salinity data at different depths, not straightforward for oceanographic interpretation. Namely, mean and min/max values of salinity at the surface and at the 60 and 100 m depth, presented in Table 1 reveal that at the surface, mean salinity is 8.18 psu (from 5.74 to 9.94), but in the depth salinity is much lower, 4.12 psu at 60 m depth and 4.79 psu at 100 m depth, with min/max range from 6.32 to 9.05 and 7.48 to 9.94 psu, respectively. This makes the vertical profile hydrostatically strongly unstable that is not in agreement with oceanographic concepts. While the mean salinity values are not understandable, the time series

analysis of trends (Fig. 9, Fig. 11) and seasonal cycle (Fig. 12) also cannot be trusted. Temperature calculations obviously have the same problems.

The sub-chapter "4.2 Trends and variability of temperature and salinity" should be fully recalculated and rewritten. Conclusions from 2 to 4 should be revised as well. To achieve meaningful information for oceanography, perhaps the revised presentation could follow some aspects of the sub-basin approach.

**D) Advancing oceanographic knowledge**

If the authors wish to focus the MS on the assessment (instead of data assimilation), the oceanographic aspect of the study has to be strengthened. The publication list is dominated by references to modelling and data assimilation studies. These topics are also dominant in the discussion; one part out of four parts is devoted to comparison with other studies on long-term variability in the Baltic Sea.

In the chapter of results, the sub-chapter "4.1 Validation of the simulation results" is clear in content and it is well written. Regarding variability in temperature and salinity of the Baltic Sea over decadal scales (from the title of the MS), sub-chapter "4.2 Trends and variability of temperature and salinity" has serious data processing problems as indicated above. After making technical corrections, the authors should consider how to advance the oceanographic knowledge of the Baltic Sea beyond the already many publications on that topic.

**Some technical comments**
1) Presently, the text is overloaded with different numbers. The sub-sub-chapters are too much as mere explanations of figures, starting "Figure 9 and Table 1 provide …", "Figure 10 shows …", "Figure 11 reveals …", "Figure 12 provides …".
2) The statement "This is the first attempt to estimate the T/S variability at various depths within the Baltic sub-basins from a reanalysis perspective" (lines 62-63) is not clear; what would it bring new/different to the previous estimates.
3) It is not directly said what are the numbers in Table 1 in terms of averaging intervals.
4) The statement in lines 340-341 is not understandable, what is new: "This indicates that the temperature variability is larger at the surface than at other depths".
5) Fig. 3, it is not clear where to find "number of profiles at the selected monitoring stations (c)"

---

## Referee Comment (RC3)

**Review of the manuscript "An assessment of the variability in temperature and salinity of the Baltic Sea from a simulation with data assimilation for the period 1990 to 2020" written by Ye Liu, Lars Axell, and Jun She, submitted to Ocean Science journal.**

The study presents a new dataset of the physical parameters of the Baltic Sea. The dataset is covering the period 1990-2020 (30 years). It is produced using a regional circulation model for the Baltic Sea (NEMO-Nordic setup) with a data assimilation (DA) technique called the local Singular Evolutive Interpolated Kalman filter (LSEIK), assimilating both in-situ T/S profiles and satellite products. To estimate the performance of the above-mentioned model setup, the study provides some validation of the results against in-situ T/S profiles, sea surface elevation from tide gauges, SST from a satellite product, and mixed layer depth (MLD) derived from in-situ observations. The validation is based on well-known quality metrics. In addition, the study investigates seasonal, inter-annual, and multidecadal dynamics of the water temperature and salinity produced by the authors' model setup at different vertical layers (surface, 60m, 100m, and bottom). By comparing model runs with and without DA, the study shows that DA improves the model quality in the Baltic Sea region. Despite some persisting uncertainties, the authors are convinced that the dataset demonstrates exemplary performance and can be utilized for further research. The study also identifies a warming trend across the Baltic Sea, with the temperatures at intermediate depths increasing faster than at the surface. As for salinity, the study highlights opposite trends in SSS across the Baltic Sea, with slight freshening in the northern part of the sea and slight salinization in the southern part. At the same time, the authors observed a notable positive trend in salinity below the halocline in the Baltic Propper. Based on the comparison with other studies and products, the authors conclude that the temperature and salinity trends are comparable to the estimates provided earlier by the community.

The study is well-structured and generally well-written. However, I have some comments on the general scientific questions it tackles. If the study's objective is solely to present and validate the new dataset, which can be inferred neither from the title nor the chosen journal, the other journal is probably the better choice (e.g., Geoscientific Model Development from the same publisher). To be published in the current journal, the authors need, in my opinion, to invest more work into the paper. First of all, I would appreciate a more detailed comparison with, e.g., the current publicly available CMEMS Baltic Sea physics reanalysis (https://doi.org/10.48670/moi-00013) covering the period from 1993 to 2023, which is the same duration as for the presented reanalysis, or possibly with other state-of-the-art Baltic Sea reanalyses (one could include it to the validation part). This analysis would demonstrate the added value of the new reanalysis described in the study.

The current analysis of temperature and salinity variability and its discussion seem to be a continuation of the reanalysis validation. In my opinion, the study, in its current form, does not deliver any new insights into T and S variability in the Baltic Sea. I think the interesting part is an investigation of T and S variability at different depths, not just at the surface, but this part needs to be improved, e.g., authors could elaborate more on the causes of the strong trends in T and S at the intermediate depths in the Baltic Propper (was it just the inflow activity or something else). In addition, some analyses and results are also currently questionable (for a detailed explanation, see the comments below). Therefore, I suggest the authors undertake a major revision and resubmit the revised manuscript. Below, I provide a few detailed comments on the manuscript:

**L21:** "*its significant impact on the Nordic climate change*": I would reformulate the sentence since the Baltic Sea is considered to be significantly driven by external forcing, not the other way round (see, e.g., Stigebrandt & Gustafsson, 2003).

**L23-24:** "*The Baltic Sea exchanges with the North Sea through the Danish Strait in the transition zone.*": Danish Strait**S**. I would also reformulate this sentence, e.g., *The water exchange between the Baltic and North Seas happens through the shallow and narrow Danish Straits.*

**L27:** *"(Matthäus et al. 2008)."*: I think a reference to newer research could be added, e.g., (Mohrholz, 2018).

**L31:** *"The Baltic Sea exhibits various features over time due to local changes in climate and forcing"*: How do the changes in climate differ from the changes in forcing? The sentence does not sound right to me; please reformulate it.

**L36-37:** *"a comprehensive climate assessment directly from the observations is still lacking due to the spatiotemporal coverage limitations, especially in the deep Baltic Sea."*: I think most observations are focused on the deep Baltic Sea, where most monitoring stations are located. Therefore, I would assume assessing the climate impact on the coastal areas is more challenging.

**L92:** "*The spin-up run (1 January 1975 – 31 December 1989)*": Could you please say a few words about the spin-up duration? I would say it is rather short.

**L119:** "*It should be note*": It should be note**D.**

**L138-139:** "*of 3 ℃ or 3 PSU, respectively*": Please comment on the thresholds. I would assume the one for salinity should be lower since its variability is lower than that of temperature.

**Eq.5:** Are you sure it should be -0.005 under exponent, not +0.005? Now, it looks like the error variances are decreasing after 70m, which contradicts the text.

**L157:** "*oN*": Typographical error.

**L170:** *"low-resolution":* In the ICES dataset, they are called "high-resolution CTD data."

**L173:** *"the SHARK and ICES data center"*. Why only those two? There is also, e.g., the IOW dataset for the Baltic Sea (https://odin2.io-warnemuende.de/).

**L177:** *"increases significantly with time."* This conclusion contradicts Fig.2, which shows a substantial gap in 2012.

**Eq.7:** Add "="

**Eq.12:** You use the standard deviation of the observations to calculate the cost function. It might affect your estimates if the observations are, for example, seasonally biased or if there are only a few profiles available. It may be better to use the model standard deviation instead. I assume the model should capture the variability more or less correctly.

**L240-241:** *"The salinity shows a stronger stratification at the eastern Gotland Basin (BY15) compared to the Bornholm Basin (BY5)."* Based on Fig.4, I would say the opposite is the case since the vertical salinity gradient seems stronger at BY5 compared with BY15. Please check.

**L242:** *"Compare"*: Compare**D.**

**L246:** *"As shown in Fig. 3, error of DA simulation is very small":* Shouldn't you refer to Fig.4 instead?

**L315-316:** *"using a seawater density criterion to define the MLD as the depth at which the seawater density deviates by 0.03 kg m-3 from the surface value (Chrysagi et al., 2021)."* I suggest adding the original publication on this method: https://doi.org/10.1029/2004JC002378. In addition, please clarify whether you used in-situ or potential density. To remove the effects of pressure, the density should be potential. But I think the difference should be very minor in the Baltic Sea.

**L322-323:** *"The MLD was shallower in the far ends of the elongated Baltic Sea, including the Bothnian Bay, the Arkona Basin, and the Gulf of Finland, compared to the MLD in the central Baltic Sea":* Is it only due to the different depths?

**L345:** *"intermedia"*: intermedia**TE.**

**L349:** *"well mixing"*: wrong use of English language.

**L352:** *"This pattern indicates a clear temporal variation of the T/S trends in the Baltic Sea.":* I would add a reference here.

**L378-379:** *"the tendency for increased salinity was more pronounced at deeper waters, as evidenced in both the Baltic Proper and the Bornholm Basin.":* Please add additional analysis on that topic.

**L381:** *"notable, significant":* a general comment. I would estimate the statistical significance of the trends presented in the paper and discuss them accordingly.

**L427:** "*On data assimilation scheme, please write*": please write.

**L431:** *"Fig. 3"*: maybe Fig. 4?

**L436:** *"Fig. 4"*: maybe Fig. 3?

**L465:** *"Fig. 3"*: maybe Fig. 2?

**L474:** "which well match with observed values,": English language

**L479:** "to accurate representation": English language

**L480:** *"data is"*: data are

**L493:** *"which is helpful to constrain the bottom simulation":* I didn't understand what the bottom simulation is.

**Table 1:** The decrease of salinity at intermediate depths looks pretty suspicious. This means that, generally, the stratification is unstable, which cannot be accurate. The same problem is visible in Fig.9. One possible explanation could be that the shallow transition region was included in the averaging at the surface but not at the intermediate depth. To avoid that and account for the different conditions in different Baltic Sea regions, I recommend providing the analysis for different basins of the Baltic Sea separately (the HELCOM definition can be applied here). In addition, I would calculate trends at all available depths and provide the vertical profiles of trends rather than focusing on specific vertical layers. If the authors still want to focus on particular depths, they should, in my opinion, account for area differences when averaging data. Are trends statistically significant?

**Figure 3:** A typo in the figure caption. There is no panel C in this figure. I suggest adding colorbar labels as well.

**Figure 4:** I suggest adding captions to the plots (station name for the profiles and an indication of whether the simulation was with or without DA for 2D plots). In addition, I would make colorbars equally spaced to facilitate the discussion of stratification, etc., in the text.

**Figure 5:** Please add labels.

**Figure 7:** Labels as well.

**Figure 8:** Also, colorbar labels. In addition, instead of "size of the mixed layer depth," I would say "mixed layer thickness."

**Figure 9:** Same as in Table 1. In addition, I would add an indication of seasonal variability on that kind of plot (e.g., +- sigma). Why is the salinity in [g kg$^{-1}$] in this figure? I suggest replacing the label with PSU since, in the model, the EOS-80 equation of state was used.

**Figure 10:** Trends significance and some discussion on observed results.

**Figure 11:** Same as in Figure 10.

**Figure 12:** Please indicate what error bars mean. This Figure also showcases the importance of the basin approach. I believe the seasonal signal in the bottom temperature is caused by disproportionally more shallow areas that enter into bottom values averages compared to the deeper regions of 60m and 100m, which, in my opinion, could lead to false conclusions.

**Literature:**

Stigebrandt, A., & Gustafsson, B. G. (2003). Response of the Baltic Sea to climate change— Theory and observations. Journal of Sea Research, 49(4), 243-256. https://doi.org/10.1016/S1385-1101(03)00021-2

Mohrholz, V. (2018). Major Baltic Inflow Statistics – Revised. Frontiers in Marine Science, 5, 385391. https://doi.org/10.3389/fmars.2018.00384

Montégut, B., Madec, G., Fischer, A. S., Lazar, A., & Iudicone, D. (2004). Mixed layer depth over the global ocean: An examination of profile data and a profile-based climatology. Journal of Geophysical Research: Oceans, 109(C12). https://doi.org/10.1029/2004JC002378

---

## Author Comment (AC2)

**Review of "An assessment of the variability in temperature and salinity of the Baltic Sea from a simulation with data assimilation for the period 1990 to 2020" by Liu et al.**

The manuscript provides a description of a model reanalysis simulation of the Baltic Sea with data assimilation for the study period 1990 to 2020. The generated data are used to study temperature and salinity trends at different depths and different sub-basins. The manuscript spends most of its content on the description, validation and discussion of the model simulation and less content on the temperature and salinity trends. From the current state of the manuscript, I do not know whether the manuscript is intended to be a model validation paper or a paper on the variability of temperature and salinity. Both parts need a lot of improvements in presentation and methods to be acceptable in the future. The trend analysis does not provide new information to the scientific community as trends have been studied a lot before. In addition, previous studies could also partially explain these trends due to changing dynamics and internal variability. This study overlooked this literature and does not add new understanding. The authors should consider splitting the manuscript into two papers in the future: one on the introduction of the model system and detailed validation of the simulation, and an improved version on T and S variability, trends and its dynamical origins, since otherwise the paper would be too long in my opinion. But the authors have to decide. Below are my main comments and suggestions on both topics.

Thank you very much for your positive and helpful comments. We have implemented all requested changes. Please find our detailed response in blue below.

**Reanalysis and validation**
-For the Baltic Sea, the saltwater inflow from the North Sea is the crucial process that renews most of the bottom water of the Baltic Sea. To see a model validation without a time series comparison of bottom salinities in the different basins is quite surprising. I would expect almost all bottom water trends to be governed by saltwater inflows. As the authors themselves state, the model does not capture this important driver of the Baltic Sea's water masses. How can the reader trust the computed trends? In order to trust the trends, a plot of the time series compared to some central stations is necessary. In general, the model should be configured in such a way that this important physical process is represented by the model's physics and not just included by data assimilation.

The model could capture key physical processes in the Baltic Sea and the North Sea. For example, the inflow/outflow water masses from the North Sea and the Baltic Sea. This has been verified by the Hordoir et al. (2019). However, the model results still include biases due to errors in initial conditions, lateral boundaries, and forcing. To address there, we use data assimilation to correct model biases and obtain a good dataset closer to the observations. For clarify, we add a time series comparison of reanalysis tempeature and salinity with observations at the BY5 and BY15 stations for both the top and bottom waters.

[Figure]

Figure 1. Surface and bottom temperature and salinity from observations and model with and without data assimilation at the BY5 and BY15 stations during 1993–2010.

-In addition, trends may be biased in cases such as the following: a monitoring station was introduced in 2000 and is used for assimilation starting in 2000. Due to the introduction of the data, the model salinity hypothetically increases by 1g/kg in the following. This would artificially create a trend, although the model's physics itself would not support a trend. Similarly, if the data assimilation were to suddenly stop during the model simulation. Do such cases occur in the simulation? If yes, the authors should be transparent about these potential biases in the trends.

The model biases are mainly due to imperfect initial conditions, which can greatly impact the accuracy and reliability of the results. As shown in Figure 1 and supported by the findings of Hordoir et al. (2019), it is clear that the model successfully captures various physical processes and represents climate feedback mechanisms effectively. This indicates that the model can simulate important Baltic dynamics, such as the inflow and outflow of water masses. However, it is important to acknowledge that model biases, arising from both forcing and initial conditions, can cause discrepancies between the modeled results and observed values. For example, the intensity of inflow and outflow events might be underestimated, or these events may be delayed within the model. To minimize these biases, data assimilation combines observational data with model predictions, producing a reliable reanalysis dataset by reducing initial biases to a small level.

It is worth noting that the data assimilation can affect the local variation of T/S. The variability and trends reported in this study emphasize overall or long-term changes, which aren't significantly affected by local adjustments of data assimilation.

Moreover, when data assimilation is stopped due to unavailability of observations, small differences between the model and observations can be maintained autonomously by the model itself. Data assimilation does not alter the internal logic or mechanisms of the model. Therefore, the bias in the presented trend statistics resulting from data assimilation can be considered negligible.

-In general, the validation lacks a validation of time series. Most validation focuses on the mean state, but as this study focuses on trends, the validation should ensure that the model can be trusted in this regard, e.g. by showing time series.

We have addressed the validation of time series in original manuscript by including a comparison of the simulations and observations in Figure 3. To further clarify, we have added another time series comparison of the observations and model with and without data assimilation at the BY5 and BY15 stations (as mentioned above). This additional comparison helps verify that the NEMO-Nordic model used in this study captures key dynamics of the Baltic Sea.

**Trends and variability**
- Regarding this part of the manuscript, all results lack a lot of critical information for the reader to trust the results, as no errors, uncertainty ranges, and significance checks are presented.

We acknowledge the importance of presenting uncertainty information for the reader's confidence in the results. All variability and trends in temperature and salinity are derived from the reanalysis dataset. To address this concern, we have validated the reanalysis and included uncertainty estimates in Figures 3–8. Additionally, we have provided uncertainty ranges in the trend statistics in the revised Table 1 and Figures 10 and 12 as followinig:

Table 1. The temperature and salinity variability in the Baltic Sea over the period 1990-2020.

| Parameter | Trend [year $^{-1}$] | Mean | Maximum | Minimum |
|-----------|----------------------|------|---------|---------|
| SST [°C] | 0.037±0.010 | 8.16±0.60 | 20.08±1.54 | 0.67±0.54 |
| T60 [°C] | 0.044±0.008 | 4.16±0.55 | 6.76±0.52 | 2.31±0.62 |
| T100 [°C] | 0.051±0.006 | 4.77±0.54 | 5.53±0.57 | 4.09±0.51 |
| SBT [°C] | 0.041 ±0.008 | 5.34±0.53 | 10.76±0.51 | 1.96±0.49 |
| SSS [PSU] | -0.004±0.002 | 5.83±0.10 | 6.50±0.10 | 5.01±0.20 |
| S60 [PSU] | 0.026±0.003 | 7.48±0.28 | 8.48±0.42 | 6.54±0.13 |
| S100 [PSU] | 0.049±0.005 | 9.16±0.51 | 9.63±0.55 | 8.61±0.51 |
| SBS [PSU] | 0.015 ±0.003 | 7.39±0.19 | 8.34±0.26 | 6.54±0.15 |

[Figure]

Figure 10. The annual mean temperature (left panel) and salinity (right panel) at surface (a,b), 60 m(c,d), 100 m (e, f), and the bottom (g, h) of the Baltic Sea for the period 1990-2020 and their linear decadal trends (i, j). Linear trends are showed by solid lines and the standard deviations are showed by the shaded area.

[Figure]

Figure 12. The monthly mean and variability of temperature and salinity of the Baltic Sea at the surface (a,b), 60 m (c,d), 100 m (e,f), and the bottom (g,h) for different decades, derived from reanalysis.

-How do the trends compare to the trends of observations over the same time period? Comparing the results with trends from other studies may be biased by the different time periods considered.

We have made the comaprison between our results and other studies over different period in the discussion section. We acknowledge the referee's point that comparing results from different time periods may introduce biases. However, these comparisons still provide us with confidence in the reliability of our study's results.

Additionally, we have compared our results with observed trends and included two recent studies on the South and North Baltic Sea—Zalewska et al. (2023) and Kankaanpää et al. (2023). These studies further verify that our results are indeed reasonable and consistent with current research.

 "To the entire Baltic Sea, this study reported a warming rate of 0.037 °C/year in the Baltic surface waters for the period 1990 to 2020, which is very similar to previous estimates from in-situ observations: 0.03–0.06 °C/year for the period from 1982 to 2016 (Liblik and Lips, 2019) and 0.028 to 0.039 °C/year in the southeastern Baltic Sea for the period 1950 to 2020 (Stockmayer and Lehmann, 2023). Further, spatial pattern of the SST trend shown in the present study is very similar to earlier studies from both model or observations. Specifically, a smaller warming trend is observed in the southern Baltic Sea, while a larger warming trend appears in the northern Baltic Sea, consistent with

the results of Siegel and Gerth (2019), Liblik and Lips (2019), and Jamali et al. (2023). Additionally, a significant warming trend is evident in the Gulf of Finland, which has been noted by Liblik and Lips (2019) and Stockmayer and Lehmann (2023).

For the salinity of the Baltic Sea, the present study reported a freshening trend of -0.004 PSU/year at the surface for the period 1990−2020, which is very similar to the previous estimations: -0.005 to -0.014 PSU/year in the upper layers for the period 1982−2016 (Liblik and Lips 2019) and -0.005 to -0.019 PSU/year in the surface waters in the northern Baltic Sea for the period 1957−2021 (Kankaanpää et al., 2023). Furthermore, this study reported a salinization trend of 0.015 to 0.049 PSU/year in the deeper waters, which is consistent with earlier estimates as well: 0.02–0.04 PSU/year in the deeper waters for the period 1982−2016 (Liblik and Lips 2019). The salinity trends are declined by 0.009 PSU/year in northern Baltic bottom in this study, which is consistent to the findings of and 0.007–0.025 PSU/year in Kankaanpää et al. (2023) for the period 1957–2021."

Zalewska, T., Wilman, B., Lapeta, B., Marosz, M., Biernacik, D., Wochna, A., Saniewski, M., Grajewska, A., Iwaniak, M. 2023. Seawater temperature changes in the southern Baltic Sea (1959–2019) forced by climate change, Oceanologia, 66, 37-55. https://doi.org/10.1016/j.oceano.2023.08.001. Its main conclusions are:

Kankaanpää, H. T., Alenius, P., Kotilainen, P., Roiha, P., 2023. Decreased surface and bottom salinity and elevated bottom temperature in the Northern Baltic Sea over the past six decades, Science of The Total Environment, 859, https://doi.org/10.1016/j.scitotenv.2022.160241.

-I do not think that the results add any new information to the Baltic Sea community. It does not add new understanding on the dynamical reasons for the well documented, observed trends which are only recreated.

We acknowledge this concern. However, the primary objective of our study is to evaluate and analyze temperature and salinity trends in the Baltic Sea from 1990 to 2020

Reply: We understand the concern. However, the primary focus of our study is not to provide new insights into the dynamics of the Baltic Sea, but rather to evaluate and analyze temperature and salinity trends in the Baltic Sea from 1990 to 2020. While our study also validates the reliability of the reanalysis dataset, the main focus remains on assessing long-term trends to ensure the dataset's accuracy for such trend analyses. We highlight our contribution to the Baltic Sea community:

- **Improvement of model predictions:** By assimilting observational data (e.g., satellite data, in-situ measurements) into model predictions, the study shows a method to improve accuracy of model precditions in the Baltic Sea by providing more accurate, reliable, and comprehensive models of the sea's dynamics. As a result, the accuracy of the presented dataset is significantly better than that of the dataset from model alone.
- **Assessing T/S trends by combining models and observations**: As Stockmayer and Lehmann (2023) highlighted, discrepancies between observed and simulated trends emphasize the importance of combining models and observations for accurate trend analysis. Most of the Baltic variability studies were focused on surface or bottom trends, and only a recent study explored the intermediate temperature trends from in-situ data (Liblik and Lips 2019) and

modeling (Dutheil et al. 2023). This study fills this gap by examining continuous temperature and salinity trends across four depths in the Baltic Sea over the past three decades. This is particularly important for the intermediate waters, where observations are sparse, model biases limit T/S changes assessments.

- **Applications and impact on future research:** With this comprehensive dataset, researchers can now analyze temperature and salinity at any depth in the Baltic Sea. Our dataset also extends predictions beyond short-term observations, making it a key reference for studies on the Baltic Sea's health and sustainability, especially in the intermediate waters.

- **Environmental implications:** Our study highlights a combination of warming and increased salinity stratification, suggesting that the Baltic Sea is experiencing more pronounced layering between surface and deep waters. This enhanced stratification may reduce oxygen transport to deeper layers, exacerbating hypoxia in deep basins and negatively impacting marine ecosystems and fish populations. Additionally, the stronger halocline could hinder nutrient mixing, potentially disrupting primary productivity and biogeochemical cycling. These changes may, in turn, affect food web dynamics and alter fish populations.

- **Policy and management recommendations:** By reliable quantifying the variability of temperature and salinty in the Baltic Sea, this study helps decision-makers in understanding how these factors influence oceanographic conditions. Our findings offer essential insights for policy development in coastal management, marine ecosystem protection, pollution control, and fisheries sustainability. Based on our results on Baltic Sea warming and salinity trends, we recommend that fishery management and coastal planning incorporate strategies for adapting to these changes, alongside enhanced monitoring of ecosystem shifts tied to warming and salinity changes.

-The whole presentation, from the introduction to the discussion to this part of the paper, is missing some critically important points, such as the large internal variability of the Baltic Sea (~30-year cycle). Due to the large natural variability in the Baltic Sea system, looking at trends over a period of similar length will most likely lead to aliasing artefacts. The authors touch on this part with their Figure 11, but do not discuss this critical property of the Baltic Sea.

We appreciate the referee for pointing out this limitation of this study. Clarifying this limitation will certainly strengthenthe study's reliability.

The Baltic Sea is affected by decadal oceanic and atmospheric cycles (e.g. Atlantic Multidecadal Oscillation (AMO) and North Atlantic Oscillation (NAO)). During positive NAO or warm AMO phases, increased atmospheric pressure gradients and stronger wind forcing enhance the influx of saltier water from deeper or surrounding areas, increasing the salinity at intermediate depths. The variability of the Baltic SST is closely linked to the AMO and NAO (Kniebusch et al., 2019). For instance, the negative phase of the AMO in the early 1990s contributed to a cooler SST trend, while the positive phases of the AMO and NAO from the mid-1990s through the 2010s contributed to a general warming trend in the Baltic Sea. Particularly, an unusual warm period in the late 2010s produced a stronger decade variability in the Baltic temperature compared to other decades. Additionally, the increasing frequency of heat waves in recent years has exacerbated warming trends in surface waters, further impacting the seasonal and interannual dynamics of T/S in the Baltic Sea.

We agree with the referee that deriving linear trends from a 30-year dataset has its limitations. In this study, we analyzed the linear trends in Baltic temperature and salinity using a 30-year reanalysis dataset. However, due to the limitations of this dataset length, it is not possible to derive long-term

trends in Baltic temperature and salinity. The trends reported in this study reflect only the variability of temperature and salinity in the Baltic Sea between 1990 and 2020 and should not be interpreted as representing variability during other periods. The T/S trends reported in this study has been verified by comparison with those from obserations. The purpose of this study is to identify trends in Baltic Sea temperature and salinity using a reliable data assimilation-based dataset, particularly for the Baltic subsurface and subbasins. Our primary focus is not on providing a comprehensive understanding of Baltic Sea dynamics.

-Many recent papers on this topic are not cited, including the Baltic Earth Assessment Report which reviewed the current state of knowledge of the Baltic Sea a few years ago.

Thank you for reminding us of these shortcomings. We have updated our references and add relevant references to further validate our conclusions and arguments. These references are sourced from the latest academic studies in this topic (Dutheil et al, 2023, Meier et al. 2023, Mohrholz, 2018, Zalewska and Lehmann 2023; Kankaanpää et al. 2023; Bittig, et al. 2024, Saraiva et al, 2019.) including the Baltic Earth Assessment Report (Meier et al. 2023).

**General and specific comments**

Introduction: Much of the recent literature is missing from the introduction. Many key points of the introduction are citing rather old papers, although newer literature with updated analyses is available.

We appreciate the referee's comment regarding the outdated references in the introduction. We have updated the references and incorporated more recent studies, including Dutheil et al, 2023, Meier et al. 2023, Mohrholz, 2018, Zalewska et al. 2023; Kankaanpää et al. 2023; Bittig, et al. 2024, Saraiva et al, 2019.

L35-37: "Although several studies have focused on long-term changes in the Baltic Sea using observations over the last two decades (Fonselius and Valderrama, 2003; Winsor et al., 2001) [...]" It does not make sense to cite papers from two-decades ago to support this statement.

We revised it with newer literatures (Zalewska et al. 2023; Kankaanpää et al. 2023; Bittig, et al. 2024)

L37: "[...] spatiotemporal coverage limitations, especially in the deep Baltic Sea." The Baltic Sea has one of the best and longest monitoring data coverage **compared** to many other marginal seas.

We agree that the Baltic Sea has one of the best and longest monitoring data coverage compared to many other marginal seas, However, the observation's coverage is still limited, especially for the subsurface waters of the Baltic Sea.

Section 2: Why are the parameters so different from the NEMO Nordic model of Kärnä et al.?

The model used in this study is NEMO-Noridic 1.0 which has a 2 nm horizontal resolution. However, Kärnä et al.(2021) used NEMO-Nordic2.0 with a 1 nm horizontal resolution.

L71-73: What kind of vertical levels do you use? I assume z*? Can you resolve the halocline and thermocline with these z-levels?

The model uses a z ∗ grid in the vertical direction consisting of 56 levels. Our model is able to capature the Baltic stratificiation and resolve its halocline and thermocline (Hordoir et al. 2019).

L78: This equation of state is outdated. Why not TEOS-10? NEMO 4.0 has it.

Our model is based on the NEMO3.6 with EOS80 state equation.

L81: How is the conversion into in-situ temperature done? With the TEOS-10 equations?

We used the EOS80 equation to do this conversion.

L88: Why do you need isopycnal diffusion to close the Neva river inflow? This seems unusual.

We followed the parameters setup used in the Hordoir et al. (2015), an additional strong isopycnal diffusion is used close to the Neva river inflow (Gulf of St. Petersburg) in order to avoid negative salinities..

L89: 3cm seems like a lot of friction. Why did you choose this high value? Other NEM Nordic setups have smaller values.

Apologies for the confusion in our previous description. The 3 cm roughness was used in a different version of the NEMO-Nordic model, as described in Hordoir et al. (2015). Our model, however, is based on the NEMO-Nordic 1.0 version from Hordoir et al. (2019), which employs a constant roughness of 1 cm. We have corrected this mistake in the revised version.

Hordoir, R., L. Axell, U. Löptien, H. Dietze, and I. Kuznetsov (2015), Influence of sea level rise on the dynamics of salt inflows in the Baltic Sea, J. Geophys. Res. Oceans, 120, doi:10.1002/2014JC010642.

L94: Why is precipitation corrected? Any validation for this change?

We compared the total precipitation from the original UERRA with the EURO4M (Euro-pean Reanalysis and Observations for Monitoring, http://www.euro4m.eu/) and obtained a multiplication factor of 0.8 for correction.

L160: What is the climatology check?

For the OSISAF SST, they have abs(SSTc-SSTclimatology) <=10 °C and abs(SSTc-SSTinsitu) <= 3.0ºC check. The product quality is also identified to 5 levels: 0: unprocessed; 1 cloudy, 2: qualitative use only: 3, 4, 5: usable data of increasing quality. We only used the quality_level >2.

L173/174: What is the source of the sea level data?

The sea level data were downloaded from CMEMS Baltic Sea - near real-time (NRT) in situ quality-controlled observations. https://doi.org/10.48670/moi-00032.

L179-183: Some typos. There are more occasions of wrong grammar and typos. I will not list them all.

We rewrite the paragraph:

"The Danish straits shows the largest number of observed T/S profiles, with 14213 profiles for temperature and 14190 profiles for salinity. In contrast, the Gulf of Riga has the lowest number of observed profiles, with only 466 temperature profiles and 457 salinity profiles. Additionally, the south Baltic Sea has more observed T/S profiles than the north Baltic Sea. Notably, there are significantly more temperature profiles than salinity profiles were observed for the period from 1990 to 2000. Furthermore, there is notable variability in the number of observed T/S profiles across CTD stations. A greater number of profiles are recorded at stations BY5, BY31, ANHE, and BY2 in comparison to other locations. As illustrated in Fig. 2c, BY5 has a largest number of observed profiles, with 956 temperature

profiles and 955 salinity profiles, while F16 has the smallest number of observed T/S profiles, with only 55 profiles. Moreover, both SR5 and LL07 has fewer than 100 observed profiles. "

The method section misses a description of trend analysis.

We added the text to describe the trend analysis method:

"In this paper, Baltic Sea mean refer to the average value for Baltic areas east of 13 °E longitude (Figure 1). The linear regression analyses were performed using the annual mean. The linear regression analysis was performed using the general least square fit method by maximum likelihood. All the trends have been significantly checked (The interpretation of significance levels based on the p value). "

Section 3.3.: Spell out the acronyms somewhere.

We checked and spelled out all the acronyms.

L209/210: Can you be 100% sure that the different databases have no overlaps and that you do not compare the reanalysis with the data used for assimilation? I do not know the databases, therefore I ask.

The database have overlaps. We compared the reanalysis with assimilated data. For example, the comparison with satellite SSTs.

[Figure]

Figure 4. The overall averaged (a) bias, (b) NRMSD of reanalyzed sea surface temperature, and (c) difference in assimilated satellite sea surface temperature relative to the CMEMS L4 sea surface temperature product for the period 1990-2020.

the differences between the DA and validation satellite SSTs could potentially impact the accuracy of the SST validation process. The discrepancy between assimilated combined satellite SST from IceMap and OSISAF and CMEMS-L4 SST varies in time and space (not shown). The CMEMS-L4 SST was warmer than both assimilated satellite SSTs and the reanalysis SST in the southwest of both the Bothnia Sea and the Bothnian Bay and the south of Gotland basin. Therefore, it is reasonable to believe that the differences between assimilated combined satellite SST and CMEMS-L4 SST contributed to the reanalyzed biases of SST in those regions relative to CMEMS-L4 SST. The opposite phenomenon occurred in the eastern Baltic Proper, the Bornholm Basin, Eastern Arkona Basin, southern Gulf of Riga, and the transition region between the North Sea and Baltic Sea (Fig. 7). These results highlight the importance of considering such discrepancies for a comprehensive assessment of the reanalysis results and their implications for understanding SST variations. The reprocessed product on multiple satellites

is better to represent a general ocean state than a single satellite product. In this study, we used a reprocessed SST (CMEMS-L4) based on multiple satellites to validate the assimilation results to verify the robustness and generalizability of the reproduced SST

L248-250: This shows that your model does not capture the essential physics of the salt water inflows, which are the central process that determines the bottom water properties. Therefore, I have little confidence in the trends of the model. Also, where does the salt come from in the DA simulation, if the model does not reproduce the physics of the inflows?

The NEMO-Nordic model used in this study is capable of capturing key Baltic dynamics, as confirmed by Hordoir et al. (2019). However, the model exhibited some bias in its simulations, primarily due to imperfect initial conditions and forcing. For instance, the initial conditions showed lower temperatures and salinities in the bottom waters. As a result, the simulated inflows were either weaker in intensity or occurred at incorrect times. By incorporating data assimilation (DA), these model biases were significantly reduced. The inflows generated through DA are now consistent with observed data, and the temperature and salinity in the bottom waters have been adjusted to levels similar to those observed.

Please also refer to the previous response on "**Reanalysis and validation**" for further details.

L258: I disagree because the model does not seem to capture one of the essential mechanisms, the saltwater inflows.

See above reply and the time series at BY15 and BY5. Our model is capable of capturing the essential Baltic dynamics like the inflows/outflows.

Section 4.1.3: SST products from satellites themselves are also imperfect.

We acknowledge that satellite-derived SST, like other satellite products, can be biased and imperfect. However, satellite SST products offer valuable insights by capturing more detailed spatial patterns and smaller-scale features of surface temperature, particularly in regions or on timescales where numerical models may face limitations in representation.

Section 4.2.: Table 1 has no uncertainties, and it is not clear how the numbers were computed. Fig. 9 also needs uncertainty range and significance checks.

We revised the figure and Table 1, we added the uncertainty in the data with standard deviation, and the trend statistics with 95% confidence interval. See our first reply to "**Trends and variability**"

L338: The cold water range is not in Tab. 1

In the revision, we used standard deviations to represent the uncertainty in temperature and salinity. As a result, we no longer mention the cold water or minimum annual mean temperature. The text has been revised to

"The temperature in the Baltic Sea also showed substantial interannual variability throughout the simulation period. For example, between 2010 and 2015, temperatures fluctuated by 0.41–1.01°C. The SD highlighted significant variability in annual mean temperatures, particularly after 2010, indicating that temperature fluctuations have been more pronounced in recent years. This suggests that warming

trends may be modulated by fluctuations in seasonal heat uptake, ice cover extent, and short-term climate variability, such as North Atlantic Oscillation (NAO). The annual temperature showed more pronounced fluctuations at the surface and bottom layers, ranged 0.67–20.08 °C and 1.96–10.76 °C, respectively. In comparison, temperature variability at intermediate waters was lower, with values of 2.31–6.76 oC at 60 m and 4.09–5.53 oC at 100 m (Table 1)."

L340: The statement is obvious.

We changed the text to "Compared to the surface waters, the subsurface waters exhibited a less variability in annual mean temperature. This indicates that the surface waters are more influenced by atmospheric variations."

L348-350: Freshest water should not be found on the bottom, except in cells where river discharge is present.

We agree with the comment and have removed the sentence: "It is also noted that both the saltiest and freshest waters occurred at the bottom."

L350-352: This result is not described in the previous text, but reads as a summary of the previous paragraphs. Again, this is not a new finding, see e.g. Kniebusch et al. (2019)

We acknowledge that Kniebusch et al. (2019) analyzed the variability of temperature and salinity trends over the period from 1850 to 2005. In our study, we focus on a more recent period, specifically highlighting the temperature and salinity variability observed between 2010 and 2015. To provide more context and clarity, we have revised the text to emphasize this more recent trend. The updated sentence now reads:

"This study shows that the temperature and SSS of the Baltic Sea showed a decreased between 1990 and 1995, followed by a increase between 2010 and 2015, reflecting clear temporal variations in the T/S trends in the Baltic Sea across the period 1990-2015. The trends observed between 2010 and 2015 indicate a shift in conditions compared to the earlier part of the period, emphasizing the dynamic and fluctuating nature of the Baltic Sea's hydrographic properties."

Section 4.2.1.: Are the trends significant? Without any significance checks, the whole section cannot be trusted.

Thanks, we added the trends significant check and hatched the high significant area (p value smaller than 0.05) and with no significant area (p value larger than 0.033).

Section 4.2.2.: The interdecadal variability of water mass properties is well described in the literature, which is overlooked by the authors, see one of my main comments.

Although the interdecadal variability of water mass properties is well-documented in the literature, as pointed out by the reviewer in their main comments, our study specifically focuses on the variability observed between 1990 and 2020. This period is the primary focus of our analysis, allowing us to highlight more recent trends in water mass properties.

L427: Incomplete sentence or paragraph

We removed this incomplete sentence "On data assimilation scheme, please write"

Figure 2: It would also be useful to see the temporal resolution of each station. Are there stations included which can provide data with very high temporal resolution, e.g. every 10 min? There exist such monitoring stations in the Baltic, e.g. the MARNET stations of the BSH in Germany

As mentioned in the study, the T/S profiles used are from SHARK and ICES data, which provide near-monthly observations, though these are spread across different stations. We did not use higher temporal resolution data, such as those available from the MARNET stations of the BSH in Germany (e.g., measurements every 10 minutes), as our primary goal was to assess long-term variability. Data with such high temporal resolution is not suitable for a 30-year simulation, as it would not be consistent with the long-period trends we aimed to capture.

Fig. 5: The color of the colorbar does not align with the colors of the dots since the dots have alpha values.

We revised it.

[Figure]

Figure 6. Global averaged bias (a,c) and RMSD (b,d) of salinity (a,b) and temperature (c,d) from reconstructed fields relative to all ICES profiles from 1990 to 2020.

Fig. 8: An additional panel with bar plots would be helpful for comparison, since the 60m range is quite large and could mask substantial MLD deviations

We agree the 60m range is large. However, a map of these stations is easier to show where we do the comparison. We revised the figure to shows both the station positions and the size of the mixed layer thickness:

[Figure]

Figure 7. Seasonal mixed layer depth for the period 1990-2020: MAM: March-April-May; JJA: June-July-August; SON: September-October-November; DJF: December-January-Feburary. The left pandel shows the station positions and the right panel shows the size of the mixed layer depth.

Fig. 9: This should include time series and trends from observations. This would help the reader to have confidence in the model results.

We appreciate this comment. We have validated our simulation results using both in-situ and satellite observations. The results show that the temperature and salinity dataset agree well with the observations. Additionally, trends derived from observations helped to verify the trends in our results. A detailed comparison of our results with those from observational data has been included in the discussion section.

Fig. 10: Indicate where the trends are statistically significant.

We added the significant check.

Fig. 11: This could also be included in Fig. 9.

We changed it.  See above response.

Fig. 12: Is this for basin averaged values? If yes, does the SD include both spatial and temporal variability? What would the same climatology look like for an observation?
It is calculated for region mean of the whole Baltic Sea. Therefore, SD include both spatial and temporal variability. We showed the climatology from observations at the figure 4.

---

## Author Comment (AC5)

**Review of the manuscript "An assessment of the variability in temperature and salinity of the Baltic Sea from a simulation with data assimilation for the period 1990 to 2020" written by Ye Liu, Lars Axell, and Jun She, submitted to Ocean Science journal.**

The study presents a new dataset of the physical parameters of the Baltic Sea. The dataset is covering the period 1990-2020 (30 years). It is produced using a regional circulation model for the Baltic Sea (NEMO-Nordic setup) with a data assimilation (DA) technique called the local Singular Evolutive Interpolated Kalman filter (LSEIK), assimilating both in-situ T/S profiles and satellite products. To estimate the performance of the above-mentioned model setup, the study provides some validation of the results against in-situ T/S profiles, sea surface elevation from tide gauges, SST from a satellite product, and mixed layer depth (MLD) derived from in-situ observations. The validation is based on well-known quality metrics. In addition, the study investigates seasonal, inter-annual, and multidecadal dynamics of the water temperature and salinity produced by the authors' model setup at different vertical layers (surface, 60m, 100m, and bottom). By comparing model runs with and without DA, the study shows that DA improves the model quality in the Baltic Sea region. Despite some persisting uncertainties, the authors are convinced that the dataset demonstrates exemplary performance and can be utilized for further research. The study also

identifies a warming trend across the Baltic Sea, with the temperatures at intermediate depths increasing faster than at the surface. As for salinity, the study highlights opposite trends in SSS across the Baltic Sea, with slight freshening in the northern part of the sea and slight salinization in the southern part. At the same time, the authors observed a notable positive trend in salinity below the halocline in the Baltic Propper. Based on the comparison with other studies and products, the authors conclude that the temperature and salinity trends are comparable to the estimates provided earlier by the community.

We thank you for your valuable comments. We have implemented all requested changes. Please find our detailed response in blue below.

The study is well-structured and generally well-written. However, I have some comments on the general scientific questions it tackles. If the study's objective is solely to present and validate the new dataset, which can be inferred neither from the title nor the chosen journal, the other journal is probably the better choice (e.g., Geoscientific Model Development from the same publisher).

The study aims to provide reliable estimates for long-term and seasonal variations of temperature and salinity (T/S) in the Baltic sub-basins during the past three decades. By utilizing the reanalysis dataset, we demonstrated a reliable and gapless variability of T/S at various depths within the Baltic sub-basins for the past three decades. Therefore, validating the reanalysis dataset is a critical component of this study, ensuring its reliability and robustness.

To be published in the current journal, the authors need, in my opinion, to invest more work into the paper. First of all, I would appreciate a more detailed comparison with, e.g., the current publicly available CMEMS Baltic Sea physics reanalysis (https://doi.org/10.48670/moi-00013) covering the period from 1993 to 2023, which is the same duration as for the presented reanalysis, or possibly with other state-of-the-art Baltic Sea reanalyses (one could include it to the validation part). This analysis would demonstrate the added value of the new reanalysis described in the study.

We added a detail comparison of this study to the CMEMS Baltic physical reanalysis product:

" Although both this reanalysis and the current CMEMS Baltic Reanalysis product (CBRP, BALTICSEA_MULTIYEAR_PHY_003_011) covered the past three decades, they differ significantly in model setup, atmospheric forcing, DA method, and assimilated satellite observation sources. Unlike this study, CBRP is based on NEMO-Nordic 2.0 with a resolution of 1 nm (Kärnä et al. 2021), as well as a coarser atmospheric forcing (ERA5 dataset with 31 km resolution), and boundary conditions from the CMEMS North West Shelf multi-year product. This study uses a multivariate EOF method to generate a sample ensemble of background error covariances, which enhances the identification of key large-scale variability patterns while minimizing noise and reducing dimensionality. This method allows for sampling various combinations of patterns from the ensemble space, enhancing the diversity of the ensemble and improving DA accuracy, leading to a more reliable simulation. Furthermore, the study assimilated the satellite observations from OSISAF SST and IceMap, whereas the CBRP used CMEMS SST observations (SST_BAL_PHY_L3S_MY_010_040). When compared to the CBRP, the sea level anomaly in this study exhibits a stronger correlation with observations, except in Visby, Talinn, Grena, and Viken. A comparison of CFs revealed both datasets have comparable salinity quality at the same Baltic stations. However, there are notable discrepancies in SBS in both the southern Bothnian Sea and the southwestern Bothnian Bay. In the southwestern Bothnian Bay, the quality of SBS reported in this study was classified as good, whereas the CBRP was classified as poor. Conversely, in the southern Bothnian Sea, the CBRP data was of higher quality than the SBS in this study. Regarding temperature, the CBRP was of high quality across all stations, while this study reported slightly lower quality in the southern Bothnian Sea, though still within acceptable limits. Lastly, all reanalysis provide valuable insights into the Baltic Sea's physical conditions. "

The current analysis of temperature and salinity variability and its discussion seem to be a continuation of the reanalysis validation. In my opinion, the study, in its current form, does not deliver any new insights into T and S variability in the Baltic Sea. I think the interesting part is an investigation of T and S variability at different depths, not just at the surface, but this part needs to be improved, e.g., authors could elaborate more on the causes of the strong trends in T and S at the intermediate depths in the Baltic Propper (was it just the inflow activity or something else). In addition, some analyses and results are also currently questionable (for a detailed explanation, see the comments below). Therefore, I suggest the authors undertake a major revision and resubmit the revised manuscript. Below, I provide a few detailed comments on the manuscript:

We greatly appreciate the referee's thoughtful feedback. While we understand that the current analysis of temperature and salinity variability may appear as an extension of the reanalysis validation, the primary goal of this study is to provide reliable estimates for long-term and seasonal variations of T/S across the Baltic Sea, particularly in the sub-basins and at various depths.

We agree that focusing on the variability of T/S at different depths—beyond just the surface—offers important insights. This is a key strength of our study, and we value the referee's suggestion to explore this aspect further. We have revised the manuscript to provide more detailed discussion and context regarding the causes of the strong trends in T/S, especially at intermediate depths in the Baltic Proper. These trends may result from a combination of factors such as inflow activity, changes in stratification, and other regional dynamics. We have added more focus on these causes in the revised manuscript.

We also acknowledge that some aspects of our analysis and results may require further clarification. We are committed to improving the manuscript in response to your suggestions. Below we highlight the key findings

of our study that directly address the study's objectives and make a significant contribution to understanding Baltic Sea dynamics:

- **Filling the knowledge gap**: Most previous Baltic Sea variability studies have focused on surface or bottom trends, with only a few addressing intermediate temperature trends (e.g., Liblik and Lips, 2019; Dutheil et al., 2023). Our study fills this gap by providing a **comprehensive, gapless analysis** of temperature and salinity across **four depths** in the Baltic Sea, with a particular focus on the **intermediate waters**. This is especially crucial for data-sparse regions where observations are limited by coverage and model biases.

- **Trends at intermediate depths**: Our analysis revealed that **temperature and salinity fluctuations have been more pronounced in recent years**, particularly at intermediate depths (e.g., at 100m). These strong trends highlight the critical role of mid-depth dynamics in shaping the Baltic Sea's response to climate and oceanographic changes, which has not been fully explored in previous studies.

- **Impact of inflows and other factors**: Our findings underscore that the **strong trends** observed at intermediate depths in **the Baltic proper** are likely driven by a combination of inflow activity, changes in water column stratification, and other local factors. We provide more detailed discussion on how these factors have contributed to the observed trends in the revised manuscript.

- **Dataset's significance**: The dataset presented in this study provides a **comprehensive reference** for analyzing temperature and salinity at **any depth** in the Baltic Sea. By extending the time period and offering gapless coverage, it enables researchers to **analyze long-term and seasonal variations** beyond short-term observations, which can help improve predictions and environmental monitoring for the Baltic Sea.

- **Environmental implications**: Our study highlights a combination of **warming and increased salinity stratification**, suggesting that the Baltic Sea is experiencing more pronounced **layering** between surface and deep waters. This enhanced stratification could have significant ecological implications, such as reducing oxygen transport to deeper layers and **exacerbating hypoxia** in deep basins. These changes could negatively impact marine **ecosystems and fish populations**, disrupt **nutrient mixing**, and alter **primary productivity** and **biogeochemical cycling**.

We hope these changes address the referee's concerns and provide a clearer emphasis on the study's contributions and findings. We are confident that the revised manuscript, with an expanded discussion of the causes of T/S trends, will enhance the understanding of variability in the Baltic Sea and make an important contribution to the field.

L21: "its significant impact on the Nordic climate change": I would reformulate the sentence since the Baltic Sea is considered to be significantly driven by external forcing, not the other way round (see, e.g., Stigebrandt & Gustafsson, 2003).

We appreciate the comment and have revised the sentence to "The variability of the Baltic Sea has received considerable attention from both the scientific and political-economic communities due to its significant response to external forcings, which play a substantial role in influencing the Nordic climate change."

L23-24: "The Baltic Sea exchanges with the North Sea through the Danish Strait in the transition zone.": Danish StraitS. I would also reformulate this sentence, e.g., The water exchange between the Baltic and North Seas happens through the shallow and narrow Danish Straits.

Thanks, we revised it.

L27: "(Matthäus et al. 2008).": I think a reference to newer research could be added, e.g.,

(Mohrholz, 2018).

We updated the reference and added Mohrholz, 2018 and Lehmann et al. 2022.

L31: "The Baltic Sea exhibits various features over time due to local changes in climate and forcing": How do the changes in climate differ from the changes in forcing? The sentence does not sound right to me; please reformulate it.

Climate changes refer to long-term atmospheric and oceanic shifts, while forcing includes external factors, both natural and human-driven, that can occur on different timescales, ranging from seasonal to decadal variations. Therefore, we revised the sentence to "The Baltic Sea exhibits various features over time due to both gradual shifts in climate and changes in external forcings."

L36-37: "a comprehensive climate assessment directly from the observations is still lacking due to the spatiotemporal coverage limitations, especially in the deep Baltic Sea.": I think most observations are focused on the deep Baltic Sea, where most monitoring stations are located. Therefore, I would assume assessing the climate impact on the coastal areas is more challenging.

While most observations are concentrated in the deeper regions of the Baltic Sea, where the majority of monitoring stations are located, these stations do not provide full spatial coverage of the entire Baltic Sea. As a result, using these observations alone to assess climate impacts in the subsurface Baltic Sea can only offer a general understanding. This approach does not capture the detailed variability in areas with sparse or no observations, limiting the comprehensiveness of the climate assessment. We revised the text:

"a comprehensive climate assessment based on the observations alone remains limited due to gaps in the spatiotemporal coverage , especially in subsurface waters of the Baltic Sea."

L92: "The spin-up run (1 January 1975 – 31 December 1989)": Could you please say a few words about the spin-up duration? I would say it is rather short.

We revised the text to

"The spin-up run (1 January 1975 – 31 December 1989) was initialized using the stable restart file from Hordoir et al., (2019), with the same model version and configuration. Although the restart file represents a stable state, the simulation was continued to ensure the system adjusts to any differences in boundary conditions or forcing, allowing the model to fully stabilize within our specific setup."

L119: "It should be note": It should be noteD.

Corrected.

L138-139: "of 3 oC or 3 PSU, respectively": Please comment on the thresholds. I would assume the one for salinity should be lower since its variability is lower than that of temperature.

These values are used to identify the differences between the model and observations, based on a rough comparison between the model (without data assimilation) and observational data. If the difference between the model and observation exceeds the thresholds, we do not assimilate that observation into the model, as large adjustments could lead to instability. It is important to note that the values for salinity and temperature may differ from one another depending on the comparison between the model and observations.

Eq.5: Are you sure it should be -0.005 under exponent, not +0.005? Now, it looks like the error variances are decreasing after 70m, which contradicts the text.

The value should be -0.005 because we assume that observation error decreases with increasing depth. This assumption allows us to assign more weight to observations relative to the model values in the data assimilation process. By doing so, we aim to adjust the model more significantly in deeper waters, where observation errors are considered smaller.

L157: "oN": Typographical error.

Corrected.

L170: "low-resolution": In the ICES dataset, they are called "high-resolution CTD data."

Most of the ICES station observations have relatively low temporal and spatial resolutions. For instance, the vertical resolution of ICES observations typically ranges from 5 to 25 meters, and the temporal resolution averages about 3 to 7 profiles per month. While certain stations in the ICES database offer higher resolution, the majority of stations still operate at lower resolutions.

L173: "the SHARK and ICES data center". Why only those two? There is also, e.g., the IOW dataset for the Baltic Sea (https://odin2.io-warnemuende.de/).

our research is constrained by the exclusion of certain available observations for assimilation. We used the SHARK and ICES data, as these two data centers contain most of the important stations in the Baltic Sea.

We point out this limitation of this study by revised the text:

"it is important to acknowledge that our research is constrained by the exclusion of certain available observations for assimilation. For example, we haven't assimilated the Baltic Sea- In Situ Near Real Time Observations ( INSITU_BAL_PHYBGCWAV_DISCRETE_MYNRT_013_032), consist of level 2 data sourced from various platforms such as ferry boxes, gliders, and Argo floats. However, the presented study has already assimilated the T/S profile observations from important Baltic mooring stations in the Baltic Sea- In Situ Near Real Time Observations. To the surface satellite data, we utilize a coarse level 2 satellite product that aligns more closely with the horizontal resolution of the model employed in this study. Further, the satellite SST assimilated in current Baltic CMEMS reanalysis serves as a benchmark for validating the reanalyzed SST of this study. "

L177: "increases significantly with time." This conclusion contradicts Fig.2, which shows a substantial gap in 2012.

We revised it to "There is a significant trend towards an increase in the number of observed profiles over time."

Eq.7: Add "="

Added.

Eq.12: You use the standard deviation of the observations to calculate the cost function. It might affect your estimates if the observations are, for example, seasonally biased or if there are only a few profiles available. It may be better to use the model standard deviation instead. I assume the model should capture the variability more or less correctly.

We used a modified cost function by Eilola et al. (2011) to calcuated the assessement values. The equation has been corrected: $C = \frac{\overline{|m-O|}}{\sigma_o}$. Therefore, the standard deviation of the observations is suitable for this calculations. We agree with you that the assessments are influenced by the observation's distribution. We have discussed this point.

L240-241: "The salinity shows a stronger stratification at the eastern Gotland Basin (BY15) compared to the Bornholm Basin (BY5)." Based on Fig.4, I would say the opposite is the case since the vertical salinity gradient seems stronger at BY5 compared with BY15. Please check.

Thanks to point out this inaccurate statement. The Baltic Sea is strongly stratified, both by salinity and temperature. Although the salinity gradient at BY5 is larger compared to that at BY15, the stratification at BY15 is still stronger than that at BY5.

To clear, we removed this sentence: "The salinity shows a stronger stratification at the eastern Gotland Basin (BY15) compared to the Bornholm Basin (BY5)."

L242: "Compare": CompareD.

Corrected.

L246: "As shown in Fig. 3, error of DA simulation is very small": Shouldn't you refer to Fig.4 instead?

Thanks, we corrected it.

L315-316: "using a seawater density criterion to define the MLD as the depth at which the seawater density deviates by 0.03 kg m$^{-3}$ from the surface value (Chrysagi et al., 2021)." I suggest adding the original publication on this method: https://doi.org/10.1029/2004JC002378. In addition, please clarify whether you used in-situ or potential density. To remove the effects of pressure, the density should be potential. But I think the difference should be very minor in the Baltic Sea.

We added the reference:

Montégut, D. B., Madec, C., G., Fischer, A. S., Lazar, A., and Iudicone, D.: Mixed layer depth over the globalocean: An examination of profile data and a profile-based climatology, J. Geophys. Res., 109, C12003, doi:10.1029/2004JC002378., 2004.

We made it clear that the density for calculation is the in-situ density.

L322-323: "The MLD was shallower in the far ends of the elongated Baltic Sea, including the Bothnian Bay, the Arkona Basin, and the Gulf of Finland, compared to the MLD in the central Baltic Sea": Is it only due to the different depths?

The MLD in the far ends of the elongated Baltic Sea is smaller than that in central Baltic Sea has been caused by various factors, such as reduced wind forcing, freshwater input from rivers, and the influence of regional atmospheric conditions, which can limit vertical mixing in these areas.

L345: "intermedia": intermediaTE.

Corrected.

L349: "well mixing": wrong use of English language.

We revised the text and removed this sentence.

L352: "This pattern indicates a clear temporal variation of the T/S trends in the Baltic Sea.": I would add a reference here.

We revised the text and added a reference:

"This study showed that the temperature and SSS of the Baltic Sea decreased between 1990 and 1995, , followed by a increase between 2010 and 2015, reflecting clear temporal variations in the T/S trends in the Baltic Sea between 1990 and 2015 (Kniebusch et al, 2019)."

L378-379: "the tendency for increased salinity was more pronounced at deeper waters, as evidenced in both the Baltic Proper and the Bornholm Basin.": Please add additional analysis on that topic.

We have revised the text and re-analysis the trends distribution.

- The warming was least pronounced at the surface, while the most significant warming occurred at mid-depths (100 m), suggesting warming is not solely driven by surface heat exchange but also by internal processes, such as stratification and reduced vertical mixing.
- The increased salinity in deeper waters of the Baltic proper is primarily due to the stratification of the water column, the limited vertical mixing between surface and deep waters, and the inflow of saltier water masses from the North Sea, which tend to sink and accumulate at greater depths.

L381: "notable, significant": a general comment. I would estimate the statistical significance of the trends presented in the paper and discuss them accordingly.

Thanks, we revised the figure and text by adding the statistical significance of the trends.

L427: "On data assimilation scheme, please write": please write.

We removed these words.

L431: "Fig. 3": maybe Fig. 4?

Corrected.

L436: "Fig. 4": maybe Fig. 3?

Corrected.

L465: "Fig. 3": maybe Fig. 2?

Corrected.

L474: "which well match with observed values,": English language

We revised it to "adding observational information through DA improves the simulation accuracy and provides a reliable distribution of T/S."

L479: "to accurate representation": English language

we revised it to "as smaller or unevenly distributed datasets might limit the model's ability to accurately represent T/S dynamics across the studied region"

L480: "data is": data are

corrected

L493: "which is helpful to constrain the bottom simulation": I didn't understand what the bottom simulation is.

We revised it to "the study introduces a more advanced approach by allowing the observation error to vary with water depth for a given profile, improving the accuracy of the bottom T/S simulation of the Baltic Sea using an ensemble DA method, such as LSEIK."

Table 1: The decrease of salinity at intermediate depths looks pretty suspicious. This means that, generally, the stratification is unstable, which cannot be accurate. The same problem is visible in Fig.9. One possible explanation could be that the shallow transition region was included in the averaging at the surface but not at the intermediate depth. To avoid that and account for the different conditions in different Baltic Sea regions, I recommend providing the analysis for different basins of the Baltic Sea separately (the HELCOM definition can be applied here). In addition, I would calculate trends at all available depths and provide the vertical profiles of trends rather than focusing on specific vertical layers. If the authors still want to focus on particular depths, they should, in my opinion, account for area differences when averaging data. Are trends statistically significant?

Thank you for your insightful comments. In the original version, we inadvertently included the shallow transition zones between the North Sea and the Baltic Sea in our calculations, which may have influenced the results. Based on your suggestion, we have now recalculated the values, restricting our analysis to the Baltic Sea region east of 13°E longitude. This adjustment ensures a more accurate representation of the Baltic Sea's conditions.

To address your concern about stratification and to improve the clarity of our analysis, we have now provided separate trend analyses for different sub-basins of the Baltic Sea, using the HELCOM definition. The trend maps are now presented at four distinct depths: the surface, two intermediate depths (shallow and deeper than the permeant thermocline/halocline), and the bottom layer. We believe these four depths offer a comprehensive representation of the Baltic Sea's vertical variability.

Additionally, we have included statistical significance checks for the trends. The updated table and figure reflecting these changes are provided below.

Table 1. The temperature and salinity variability in the Baltic Sea over the period 1990-2020.

| Parameter | Trend [year $^{-1}$] | Mean | Maximum | Minimum |
|---|---|---|---|---|
| SST [°C] | 0.037±0.010 | 8.16±0.60 | 20.08±1.54 | 0.67±0.54 |
| T60 [°C] | 0.044±0.008 | 4.16±0.55 | 6.76±0.52 | 2.31±0.62 |
| T100 [°C] | 0.051±0.006 | 4.77±0.54 | 5.53±0.57 | 4.09±0.51 |
| SBT [°C] | 0.041 ±0.008 | 5.34±0.53 | 10.76±0.51 | 1.96±0.49 |

| | | | | |
|---|---|---|---|---|
| SSS [PSU] | -0.004±0.002 | 5.83±0.10 | 6.50±0.10 | 5.01±0.20 |
| S60 [PSU] | 0.026±0.003 | 7.48±0.28 | 8.48±0.42 | 6.54±0.13 |
| S100 [PSU] | 0.049±0.005 | 9.16±0.51 | 9.63±0.55 | 8.61±0.51 |
| SBS [PSU] | 0.015 ±0.003 | 7.39±0.19 | 8.34±0.26 | 6.54±0.15 |

[Figure]

Figure 10. The annual mean temperature (left panel) and salinity (right panel) at surface (a,b), 60 m(c,d), 100 m (e, f), and the bottom (g, h) of the Baltic Sea for the period 1990-2020 and their linear decadal trends (i, j). Linear trends are showed by solid lines and the standard deviations are showed by the shaded area.

Figure 3: A typo in the figure caption. There is no panel C in this figure. I suggest adding colorbar labels as well.

We added labels for Panel C and the yaxis in Figure 3 shows the size of the bars. Thanks for your suggestion. Since the yaxis shows the size of the observations and the grid lines make it easy to identify their size. The color bar is not very useful for identifying the size of the observations. the figure is revised to

[Figure]

Annual number of profiles (a), total number of profiles in the Baltic sub-basins (b), and number of profiles at the selected stations (c) for the period 1990-2020.

Figure 4: I suggest adding captions to the plots (station name for the profiles and an indication of whether the simulation was with or without DA for 2D plots). In addition, I would make colorbars equally spaced to facilitate the discussion of stratification, etc., in the text.

Thanks for your suggestion, we revised the figure and added the captions:

[Figure]

Monthly, seasonal and period averages of salinity and temperature at BY5 (a-f) and BY15 (g-l) for the period 1990-2020. Time averages (a, d, g, j) are shown over the entire water column and the standard deviation of observations over the period are shown as a grey shaded area. The seasonal variable simulated from model with (c,f,i,l) and without (b,e,h,k) data assimilation are compared with the observational monthly average values (the filled circles).

Figure 5: Please add labels.

Added.

Figure 7: Labels as well.

Labeled.

Figure 8: Also, colorbar labels. In addition, instead of "size of the mixed layer depth," I would say "mixed layer thickness."

We revised this figure.

[Figure]

Fig 9. Seasonal mixed layer depth for the period 1990-2020: MAM: March-April-May; JJA: June-July-August; SON: September-October-November; DJF: December-January-Feburary. The left panel shows the station positions and the right panel shows the size of the mixed layer thickness.

Figure 9: Same as in Table 1. In addition, I would add an indication of seasonal variability on that kind of plot (e.g., +- sigma). Why is the salinity in [g kg-1] in this figure? I suggest replacing the label with PSU since, in the model, the EOS-80 equation of state was used.

We replaced the label with PSU and added the variability.

Figure 10: Trends significance and some discussion on observed results.

We revised it to a significant marked figure.

[Figure]

Figure 11. Linear trends in the water temperature and salinity in the Baltic Sea during the period 1990-2020, derived from the reanalysis. The Areas with a significance level are hatched (p values lower than 0.05), and areas with no significance (p values larger than 0.33) are excluded (white).